# OmniSpatial: Towards Comprehensive Spatial Reasoning Benchmark for Vision Language Models

**Mengdi Jia**[1*]    **Zekun Qi**[14*]    **Shaochen Zhang**[2]    **Wenyao Zhang**[34]
**Xinqiang Yu**[4]    **Jiawei He**[4]    **He Wang**[45‡]    **Li Yi**[16‡]

[1]Tsinghua University    [2]Xian Jiaotong University    [3]Shanghai Jiao Tong University
[4]Galbot    [5]Peking University    [6]Shanghai Qi Zhi Institute

🏠 Project Page       Evaluation Code       🤗 Dataset

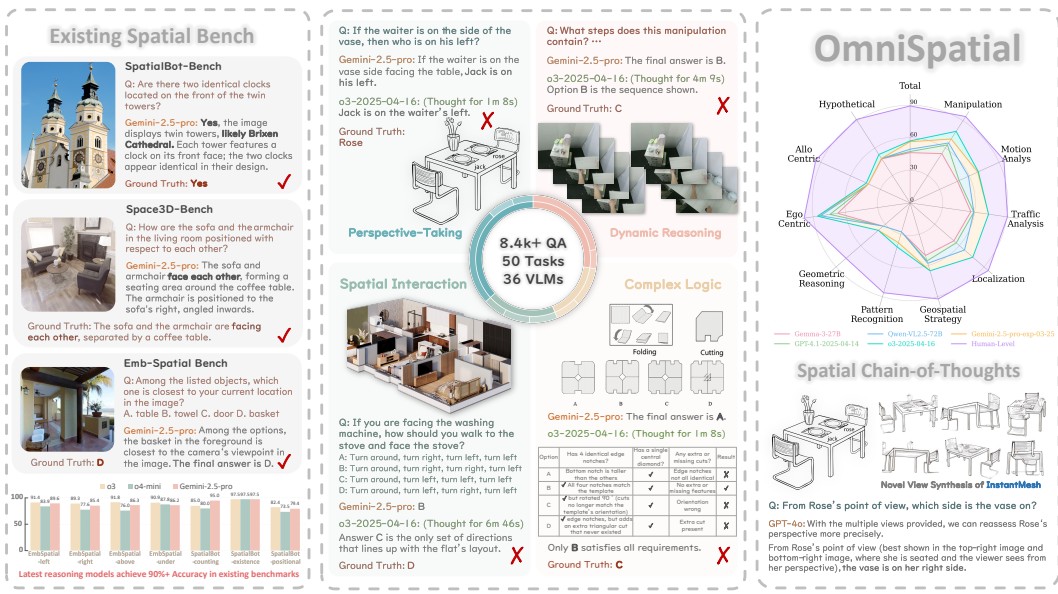

Figure 1: **Overview of OmniSpatial Benchmark.**

## Abstract

Spatial reasoning is a key aspect of cognitive psychology and remains a bottleneck for current vision-language models (VLMs). While extensive research has aimed to evaluate or improve VLMs' understanding of basic spatial relations, such as distinguishing left from right, near from far, and object counting, these tasks cover only the most elementary layer of spatial reasoning and are largely approaching saturation in the latest reasoning models. In this work, we introduce OmniSpatial, a comprehensive and challenging benchmark for spatial reasoning, grounded in cognitive psychology. OmniSpatial covers four major categories: dynamic reasoning, complex spatial logic, spatial interaction, and perspective-taking, with 50 fine-grained subcategories. Through careful manual annotation, we construct over 8.4K question-answer pairs. Extensive experiments show that both open- and closed-source VLMs exhibit significant limitations in comprehensive spatial reasoning. We also explore two strategies—PointGraph (explicit scene graph cues) and SpatialCoT (novel-view chain-of-thought)—to bolster spatial reasoning.

---

*Equal contribution. ‡Corresponding author.

# 1    INTRODUCTION

Spatial reasoning plays a crucial role in bridging visual observation to robotic action (Huang et al., 2024b; Qi et al., 2024; 2025), autonomous driving, and AR/VR. For models to execute tasks effectively, they must understand spatial relationships to determine appropriate actions. To enhance the spatial understanding of Vision-Language Models, prior works (Chen et al., 2024; Cheng et al., 2024; Cai et al., 2024; Ma et al., 2024; Song et al., 2024; Yuan et al., 2024; Yang et al., 2024; Qi et al., 2025) have integrated spatial information into datasets, enabling basic forms of spatial reasoning. Various benchmarks (Du et al., 2024; Szymanska et al., 2024; Shiri et al., 2024; Fu et al., 2024b; Song et al., 2024; Yang et al., 2024) have been introduced to systematically evaluate such capabilities, focusing on tasks like recognizing left and right, estimating depth, and constructing cognitive maps (Yang et al., 2024; Tolman, 1948; Momennejad et al., 2023). Additionally, spatial reasoning has been applied to manipulation tasks (Driess et al., 2023; Qi et al., 2025; Yuan et al., 2024), allowing systems to position objects according to specified spatial rules.

However, existing benchmarks still target basic spatial understanding, such as position relationship (left, right, front, back), proximity (near, far), and object counting. We use the latest reasoning models and agents to evaluate these benchmarks, such as o3 (OpenAI, 2025b) and Gemini-2.5-Pro (Reid et al., 2024). The results are shown in the lower left of Fig. 1. These models **have achieved over 90% accuracy on previous benchmarks** such as SpatialBot-Bench (Cai et al., 2024) and EmbSpatial (Du et al., 2024), suggesting that these basic tasks are approaching saturation.

We believe complex spatial reasoning remains a significant challenge (Gardner, 2011; Baddeley, 1998; Previc, 1998; Kosoy et al., 2025; Pothiraj et al., 2025; Chen et al., 2025a). Human interaction with the physical world often involves interpreting ambiguous, dynamic, and context-dependent spatial relationships (Bar-Anan et al., 2006; Trope & Liberman, 2010; Ramalho et al., 2018). For example, in an emergency, knowing that an AED is "to the right of the door" is insufficient without understanding schematic diagrams, correlating maps with real-world environments, and planning an efficient route. Similarly, tasks like inserting a knife into a rack or flattening a box demand reasoning about object rotation, deformation, and spatial compatibility—far beyond static object placement.

From the perspective of cognitive psychology, complex spatial reasoning goes beyond simple relational judgments, encompassing dynamic world-knowledge reasoning, interactive spatial behavior with environments or agents, logical analysis of three-dimensional structures, and perspective-taking abilities. Motivated by these challenges, we introduce OmniSpatial, a comprehensive benchmark designed to capture the breadth and depth of spatial cognition. OmniSpatial systematically categorizes spatial reasoning into four core dimensions—dynamic reasoning, complex spatial logic, spatial interaction, and perspective-taking—thus providing a principled foundation for developing next-generation spatially- and physically-aware AI systems.

Table 1: **Comparison with other spatial reasoning benchmarks**. A comparison between OmniSpatial and other existing spatial reasoning benchmarks. OmniSpatial avoids template-based annotations, features highly diverse data, and includes a significantly larger number of tasks.

| Dataset | Embodied | Task Categories | Data Domain | Data Annotation | Data Scale | Spatial QAs |
|---|:---:|:---:|:---:|:---:|:---:|:---:|
| EmbSpatial-Bench (Du et al., 2024) | ✓ | 6 | Indoor (ScanNet, *etc.*) | Template | 2.2K | 3.6K |
| Space3D-Bench (Szymanska et al., 2024) | ✗ | 6 | Indoor (Replica) | Manual | 211 | 1K |
| Visual Spatial (Liu et al., 2023a) | ✗ | 7 | MSCOCO | Template | 10K | 10K |
| SpatialRGPT-Bench (Cheng et al., 2024) | ✗ | 12 | Urban, Indoor, Sim | Template | 1.4K | 1.4K |
| What's up (Kamath et al., 2023) | ✗ | 6 | Household, GQA, COCO | Template | 5K | 5K |
| Spatial-MM (Shiri et al., 2024) | ✗ | 4 | Internet | Template | 2.3K | 2.3K |
| RoboSpatial (Song et al., 2024) | ✓ | 4 | Indoor, tabletop | Template | 1M | 3M |
| SpatialVLM (Chen et al., 2024) | ✗ | 2 | WebLi | Template | 546 | 546 |
| SpatialBot-Bench (Cai et al., 2024) | ✓ | 5 | COCO,VG,RTX | Manual | 200 | 360 |
| VSI-Bench (Yang et al., 2024) | ✓ | 8 | Indoor | Template | 288 | 5K |
| OmniSpatial (Ours) | ✓ | **50** | Internet | Manual | 6.5K | 8.4K |

The OmniSpatial benchmark includes images or video frames across diverse scenes, resolutions, lighting conditions, and weather patterns, collected from multiple countries across different continents. We evaluate state-of-the-art VLMs on our benchmark. Our findings indicate that, while current models perform well on conventional benchmarks, OmniSpatial presents a significantly greater challenge due to its comprehensive and complex task design.

Our key contributions are as follows:

- We categorize visual-spatial reasoning into four key dimensions—dynamic reasoning, complex spatial logic, spatial interaction, and perspective-taking—broadening the scope of evaluation and guiding future research on spatial cognition in embodied & physics intelligence.

- We develop the OmniSpatial dataset, which offers a diverse and challenging set of spatial tasks, serving as a comprehensive benchmark for assessing VLMs' spatial reasoning capabilities.

- We explore the enhancement of spatial reasoning in VLMs by incorporating auxiliary models in a chain-of-thought manner, demonstrating improved reasoning performance through this approach.

## 2 PRELIMINARIES: VISUAL–SPATIAL REASONING

Spatial reasoning constitutes the cognitive bridge between visual perception and geometric understanding. We define *visual–spatial reasoning* as the capacity of an artificial system to **infer, predict, and reason spatial properties of the world from visual observations**. Formally, let an RGB observation stream be $\mathbf{I}_{1:T}$ and a task-specific query be $q$. A model possesses visual–spatial reasoning that learn a mapping:

$$f : (\mathbf{I}_{1:T},\, q) \;\longrightarrow\; a, \qquad (1)$$

where $a$ belongs to a well-defined action or answer space whose correctness can be verified in the physical or simulated environment. This definition excludes non-visual priors so that improvements can be attributed to visual reasoning itself, yet it remains compatible with multi-modal extensions discussed in §1.

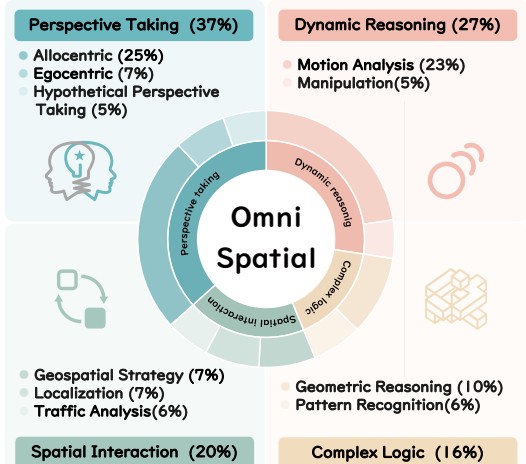

Figure 2: **Benchmark Statistics of OmniSpatial**: The distribution of tasks across 4 main categories.

### 2.1 TAXONOMY OF VISUAL–SPATIAL REASONING

Our taxonomy is motivated by two complementary perspectives. **(i) Cognitive psychology foundations.** Prior research on spatial cognition highlights partly independent faculties—such as visualization, mental rotation, perspective taking, and spatial updating—that can be systematically assessed in humans (Chabris et al., 2006; Meneghetti et al., 2022). These constructs provide a principled scaffold for analyzing how agents perceive, reason about, and act within space. **(ii) Moving beyond** *basic* **spatial relations.** Existing benchmarks are nearly saturated on simple tasks like left–right discrimination, front–back identification, and object counting (Chen et al., 2024; Cai et al., 2024; Du et al., 2024; Szymanska et al., 2024). Yet real-world embodied tasks demand richer reasoning (Kosoy et al., 2025; Pothiraj et al., 2025; Chen et al., 2025a; Stogiannidis et al., 2025; Lee et al., 2025) about scene dynamics, multi-step logic, physical interaction, and viewpoint transformation.

Guided by these considerations, we partition visual–spatial reasoning into four complementary dimensions: *dynamic reasoning, complex spatial logic, spatial interaction*, and *perspective taking*. Each dimension corresponds to a specific cognitive faculty and targets under-explored challenges in prior work, enabling us to probe a broader spectrum of spatial cognition while remaining grounded in psychological theory.

**Dynamic Reasoning** concerns inferring motion and temporal change from visual evidence. While our benchmark primarily uses static or sparsely sampled frames, such inference is crucial for adaptive decision-making in domains like robotics and navigation.

**Complex Spatial Logic** involves higher-order reasoning about relations, transformations, and geometric structures. It underpins problem-solving in design, engineering, and manipulation, where anticipating structural or relational changes is essential.

**Spatial Interaction** emphasizes reasoning guided by environmental constraints and task goals, covering skills such as path planning, obstacle avoidance, and context-aware action selection.

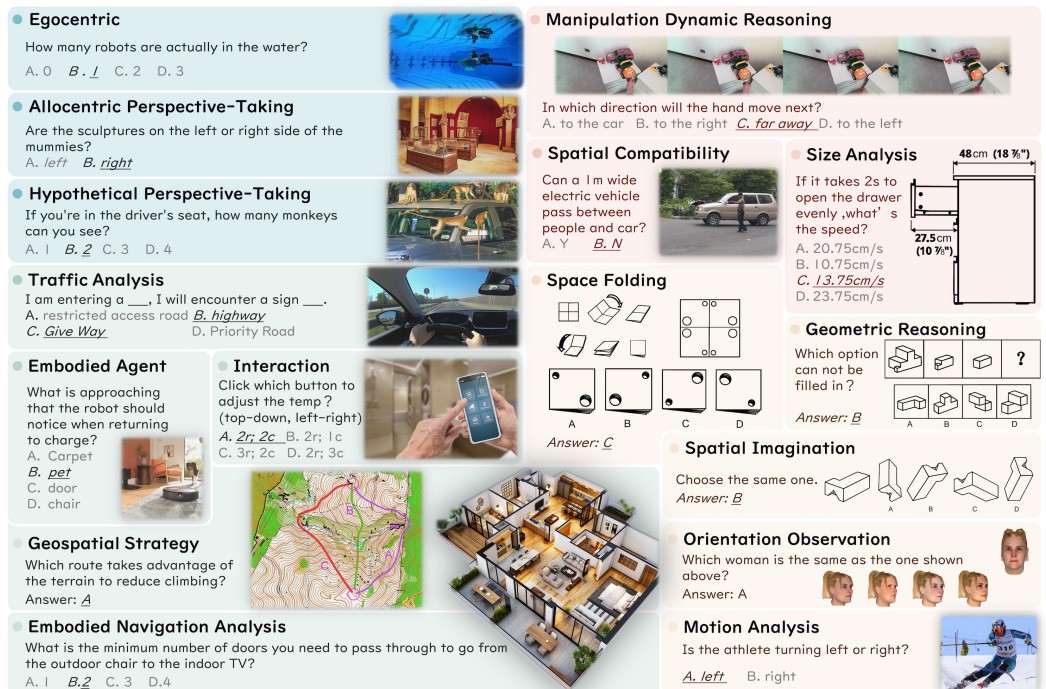

Figure 3: **Tasks Demonstration of OmniSpatial**. Several representative subtasks are selected for demonstration in each category. Note that the questions above are slightly simplified for clarity.

**Perspective Taking** captures the ability to adopt alternative viewpoints, supporting navigation, social cognition, and multi-agent coordination. It enables understanding relations from diverse perspectives and fosters flexible problem-solving.

## 2.2 RATIONALE FOR CLASSIFICATION

This taxonomy balances theoretical comprehensiveness with practical applicability. Dynamic reasoning highlights motion inference, complex logic captures abstract transformations, spatial interaction addresses real-time engagement, and perspective taking reflects cognitive flexibility. Together, these dimensions provide a framework for evaluating spatial reasoning in AI, robotics, and human cognition.

## 3 OMNISPATIAL: COMPREHENSIVE SPATIAL REASONING BENCHMARK

### 3.1 OVERVIEW

We present **OmniSpatial**, a comprehensive benchmark designed to evaluate VLMs on spatial reasoning. Rather than pursuing sheer data volume, OmniSpatial emphasizes *diversity, structure, and rigor*. It now consists of **8.4K carefully curated QA pairs**, substantially larger than earlier prototypes, and covers a broad spectrum of scenarios that demand reasoning beyond pattern recognition.

The dataset integrates heterogeneous sources—web imagery, standardized cognitive tests, driving-exam questions, and prior dataset images such as MME (Liang et al., 2024) and HOI4D (Liu et al., 2022). This mixture enriches both realism and complexity: natural images capture everyday environments and architectures; psychology-inspired tasks introduce scientifically grounded challenges; driving exams provide safety-critical dynamic reasoning; and embodied datasets contribute varied resolutions, viewpoints, and human–object interactions.

Tasks are organized into **4 major categories** and **50 fine-grained subtypes** as shown in Figs. 3 and 7, spanning from basic perspective-taking to dynamic motion prediction and spatial interaction in cluttered scenes. Each item is manually designed and reviewed through multi-round annotation, ensuring accuracy, consistency, and minimal ambiguity. This careful curation yields a benchmark

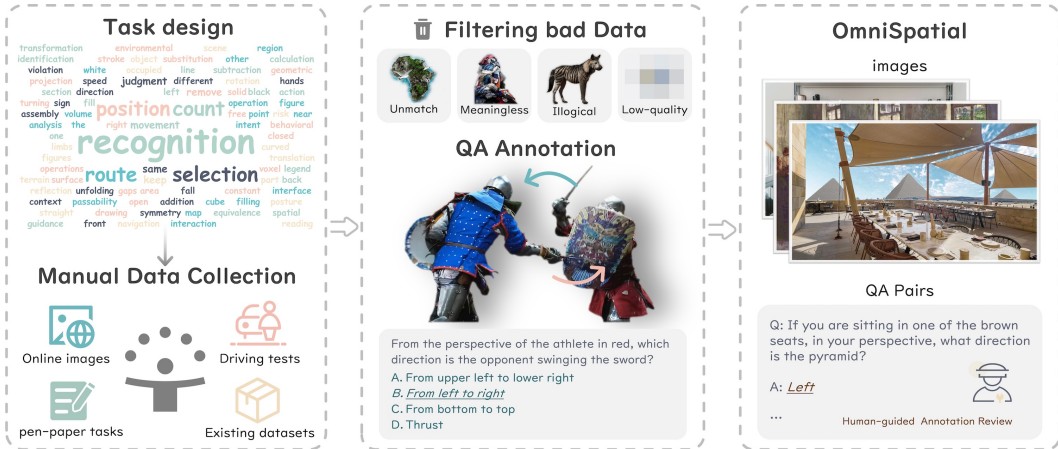

Figure 4: **Data Construction of OmniSpatial**.The pipeline collects images from multiple sources and ensures their quality and relevance through manual selection. Precise annotations are then applied, ensuring each question has a clear, unique answer while maintaining a natural, conversational expression. This process supports the effective training of VLMs in spatial reasoning tasks.

that not only broadens the coverage of spatial reasoning but also establishes a reliable ground for evaluating and advancing future multimodal intelligence.

## 3.2 BENCHMARK CONSTRUCTION

### 3.2.1 DATA COLLECTION

As described in Section 2, we define four spatial categories and corresponding task types. To build a diverse and information-rich dataset, we design targeted search strategies for each type, optimizing for relevance, diversity, and complexity.

**Web Images.** Web images form a major part of our data, especially for perspective-taking, dynamic reasoning, and spatial interaction. We use task-specific search terms (e.g., "indoor layout," "furniture arrangement") and append filters ("-ai," "-generated") to reduce synthetic content. Images are retrieved via Google's Custom Search JSON API, Web RPA, and manual search, followed by strict filtering to remove irrelevant, low-resolution, or spatially trivial cases (e.g., isolated static objects). The resulting set balances realism and complexity, ensuring broad task coverage. All images are under MIT or CC-BY 4.0 license.

**Exam-Based Test Questions.** To capture abstract spatial logic, such as 3D transformations, rotations, and perspective shifts, we collect public spatial cognition tests through web scraping and manual curation. We categorize questions by focus and difficulty to maintain balance, removing redundant or knowledge-heavy items in favor of those targeting pure spatial reasoning. This refinement increases challenge diversity and improves benchmark quality.

**Driving Test Questions.** To evaluate reasoning in dynamic environments, we source tasks from three channels: (i) image-based multiple-choice questions from driving exam websites across at least three countries, (ii) online banks of standardized tasks like turning, lane changing, and parking, and (iii) interactive U.S. driving test videos, from which we extract frames, annotate bounding boxes, and design contextual queries (e.g., "Which bounding box indicates a potential traffic hazard?"). This combination yields realistic and challenging traffic scenarios, enhancing VLM adaptability to safety-critical reasoning tasks.

**Existing Dataset Images** We integrate two key data sources: MME (Liang et al., 2024) and HOI4D (Liu et al., 2022). MME provides RGB-D data, allowing depth-based spatial inference. We leverage its depth information and **manually** propose physics-based questions such as "If a red car passes me in 5 seconds, what speed should I maintain?" This ensures realistic distance and motion-based reasoning. HOI4D contains extensive human-object interaction videos. We extract sequential frames to create motion prediction tasks, such as "Where will the hand holding the kettle move next?" By incorporating these datasets, we introduce real-world motion and interaction complexities,

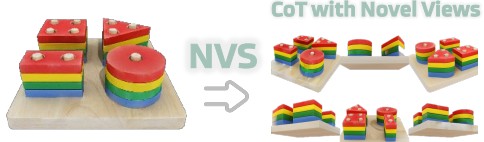

Figure 5: **PointGraph**: Enhance Spatial Reasoning through Additional Scene Graphs.

Figure 6: **SpatialCoT**: Enhancing Spatial Imagination through Novel View Synthesis.

further strengthening VLMs' dynamic reasoning capabilities. Furthermore, to extend our dataset to model training, we partition the dataset into a 1.5K test set and a 6.9K training set. The **test set is entirely human-annotated** and the train set further integrates samples from several existing datasets, including SpatialViz (Wang et al., 2025), PhysBench (Chow et al., 2025), ViewSpatial (Li et al., 2025), and DrivingVQA (Corbière et al., 2025), which substantially enhance the diversity.

### 3.2.2 QUESTION-ANSWER ANNOTATION

We design multiple-choice questions, including binary (true/false) and four-option formats, to enable standardized evaluation while minimizing annotation bias. To ensure naturalness, questions are phrased in conversational and context-rich styles (e.g., "If you are entering the classroom, on which side are the students?") rather than rigid templates (e.g., "Is [A] on the right of [B]?"). This encourages models to rely on contextual and relational reasoning rather than memorized patterns.

To guarantee clarity and answer uniqueness, we perform multiple rounds of validation and resolve ambiguous references through cross-checking among annotators. Six trained annotators—initially the authors—were involved in labeling. Each annotator underwent task-specific training, and annotations were cross-validated to reduce subjectivity. The inter-annotator agreement reached Krippendorff's $\alpha = 0.84$, indicating a high level of consistency.

### 3.3 IMPROVING VISUAL SPATIAL REASONING ABILITIES

### 3.3.1 POINTGRAPH: EXPLICIT MODELING OF OBJECT RELATIONSHIPS

To strengthen models' understanding of spatial relations, we introduce **PointGraph**, which constructs structured representations of object relationships within an image. Concretely, we employ open-vocabulary grounding models such as Florence-2 to localize multiple objects and extract their centers and bounding boxes. These detections are then assembled into a JSON-style scene graph encoding object identities and relative positions. By concatenating this structured spatial description with the original query, VLMs are provided with explicit geometric cues that facilitate more accurate reasoning about distances, directions, and configurations (see Fig. 5).

### 3.3.2 SPATIALCOT: STIMULATING SPATIAL IMAGINATION VIA NOVEL VIEWS

Human spatial reasoning often relies on mental imagery—the ability to imagine how a scene looks from different viewpoints. Inspired by this, we design **SpatialCoT**, which augments visual inputs with 3D novel views to enrich spatial imagination (Shi et al., 2023; Qi et al., 2023b; Liu et al., 2024d). Specifically, we adopt InstantMesh (Xu et al., 2024a) to synthesize six additional perspectives for each input image and compose them into a multi-view collage. This collage is then fed, along with the question, into the VLM as part of a chain-of-thought prompting pipeline. The additional perspectives provide strong geometric priors, helping models disambiguate occlusions, perspective-taking, and other view-dependent reasoning tasks.

## 4 EXPERIMENTS

### 4.1 EVALUATION SETUP

To systematically assess spatial reasoning, we evaluate models on the OmniSpatial benchmark under a unified protocol. Our evaluation covers both proprietary and open-source systems, spanning general-

Table 2: **Evaluation on OmniSpatial-test**. All models were tested 5 times and averaged to reduce randomness. **Dark green** indicates the best result and light green indicates the second-best result within the group. Gemini-2.5-Pro (Reid et al., 2024), InternVL3-78B (Zhu et al., 2025), and SoFar (Qi et al., 2025) achieve the best performance within their respective groups.

| Method | Avg. | Rank | Dynamic Reasoning | | | Spatial Interaction | | Complex Logic | | Perspective Taking | | |
|---|---|---|---|---|---|---|---|---|---|---|---|---|
| | | | Manipulate | Motion Analysis | Traffic Analysis | Locate | Geospatial Strategy | Pattern Recognition | Geometric Reasoning | Ego Centric | Allo Centric | Hypothetical |
| **Blind Evaluation** | | | | | | | | | | | | |
| Random Choice | 24.98 | - | 24.86 | 26.30 | 25.88 | 23.43 | 27.27 | 21.44 | 24.77 | 22.55 | 24.84 | 25.78 |
| GPT-3.5-turbo (Roumeliotis & Tselikas, 2023) | 30.67 | - | 38.38 | 29.19 | 38.35 | 28.76 | 36.91 | 0.82 | 24.00 | 42.16 | 33.67 | 35.90 |
| GPT-4-turbo (OpenAI, 2023) | 34.06 | - | 42.97 | 37.40 | 41.18 | 28.95 | 40.00 | 22.27 | 26.32 | 31.37 | 33.99 | 35.42 |
| **Proprietary Models** | | | | | | | | | | | | |
| GPT-4o-mini-2024-07-18 (Hurst et al., 2024) | 42.64 | 8 | 55.95 | 50.29 | 54.59 | 43.43 | 44.91 | 22.47 | 29.42 | 61.57 | 36.76 | 34.22 |
| GPT-4o-2024-11-20 (Hurst et al., 2024) | 47.81 | 5 | 65.54 | 57.23 | 56.47 | 52.38 | 54.09 | 26.29 | 25.48 | 75.98 | 39.49 | 39.76 |
| GPT-4.1-nano-2025-04-14 (OpenAI, 2025a) | 42.62 | 9 | 50.90 | 53.85 | 54.90 | 40.95 | 42.42 | 24.40 | 30.11 | 53.59 | 37.23 | 33.73 |
| GPT-4.1-mini-2025-04-14 (OpenAI, 2025a) | 48.87 | 4 | 64.32 | 56.53 | 59.06 | 60.19 | 56.36 | 29.28 | 30.19 | 72.55 | 39.57 | 39.28 |
| GPT-4.1-2025-04-14 (OpenAI, 2025a) | 51.78 | 2 | 66.22 | 64.74 | 60.00 | 65.33 | 60.18 | 31.75 | 30.06 | 70.98 | 40.64 | 39.04 |
| Claude-3-5-sonnet-20241022 (Anthropic, 2024) | 46.86 | 6 | 54.05 | 54.57 | 58.12 | 68.38 | 53.09 | 26.60 | 31.74 | 70.00 | 34.79 | 39.52 |
| Claude-3-7-sonnet-20250219 (Anthropic, 2024) | 47.53 | 5 | 57.57 | 55.95 | 56.71 | 63.81 | 59.09 | 29.48 | 28.39 | 72.16 | 36.06 | 36.63 |
| Gemini-2.0-flash-lite-02-05 (Anil et al., 2023) | 44.03 | 8 | 59.19 | 46.71 | 60.24 | 49.52 | 53.27 | 21.65 | 31.23 | 66.47 | 36.81 | 38.80 |
| Gemini-2.0-flash-exp (Anil et al., 2023) | 48.40 | 2 | 61.89 | 56.01 | 51.76 | 63.43 | 59.09 | 20.82 | 33.81 | 72.75 | 39.20 | 39.28 |
| Gemini-2.5-flash-preview-05-20 (Anil et al., 2023) | 52.12 | 1 | 67.57 | 62.72 | 68.24 | 73.33 | 60.91 | 38.14 | 34.19 | 75.49 | 35.90 | 33.73 |
| **Reasoning Models** | | | | | | | | | | | | |
| o1-2024-12-17 (Jaech et al., 2024) | 50.36 | 6 | 71.62 | 66.18 | 57.65 | 63.81 | 60.00 | 39.18 | 27.10 | 71.57 | 38.03 | 36.14 |
| o4-mini-2025-04-16 (OpenAI, 2025b) | 52.77 | 3 | 72.97 | 59.83 | 60.00 | 73.33 | 61.82 | 34.02 | 36.77 | 73.53 | 40.69 | 40.96 |
| o3-2025-04-16 (OpenAI, 2025b) | 56.33 | 1 | 71.89 | 66.18 | 61.18 | 68.57 | 65.45 | 40.21 | 29.68 | 77.06 | 48.40 | 48.19 |
| Claude-3-7-sonnet-20250219-thinking (Anthropic, 2024) | 48.62 | 7 | 57.21 | 59.73 | 53.73 | 67.94 | 57.27 | 30.24 | 28.17 | 68.63 | 37.94 | 36.95 |
| Gemini-2.5-flash-05-20-thinking (Anil et al., 2023) | 53.16 | 3 | 70.27 | 64.74 | 61.18 | 72.38 | 58.18 | 35.05 | 36.13 | 74.12 | 40.96 | 32.53 |
| Gemini-2.5-pro-preview-05-06 (Anil et al., 2023) | 55.19 | 2 | 67.57 | 71.39 | 62.35 | 75.24 | 64.55 | 43.30 | 34.84 | 74.51 | 38.03 | 37.35 |
| **Open-source Models** | | | | | | | | | | | | |
| LLavA-1.5-vicuna-7B (Liu et al., 2024c) | 34.97 | 15 | 54.46 | 31.23 | 35.29 | 36.19 | 33.94 | 29.01 | 24.18 | 55.60 | 34.66 | 36.14 |
| LLaVA-onevision-qwen2-7B (Li et al., 2024a) | 35.68 | 14 | 43.24 | 38.15 | 32.94 | 29.52 | 41.82 | 28.87 | 22.58 | 47.06 | 36.17 | 37.35 |
| LLaVA-onevision-qwen2-72B (Li et al., 2024a) | 45.66 | 6 | 62.16 | 50.29 | 54.12 | 60.95 | 56.36 | 22.68 | 25.81 | 76.47 | 37.23 | 33.73 |
| Gemma-3-4B (Kamath et al., 2025) | 39.79 | 11 | 41.89 | 49.71 | 56.47 | 27.62 | 36.36 | 23.71 | 24.52 | 59.80 | 36.17 | 38.55 |
| Gemma-3-12B (Kamath et al., 2025) | 43.71 | 8 | 54.05 | 54.91 | 54.12 | 47.62 | 45.45 | 16.49 | 30.32 | 63.73 | 36.70 | 33.73 |
| Gemma-3-27B (Kamath et al., 2025) | 44.75 | 7 | 56.76 | 55.78 | 57.65 | 50.48 | 52.73 | 27.84 | 29.03 | 64.71 | 33.51 | 32.53 |
| InternVL3-2B (Zhu et al., 2025) | 37.98 | 13 | 50.00 | 40.58 | 43.29 | 40.00 | 40.55 | 21.86 | 28.52 | 55.49 | 35.11 | 33.01 |
| InternVL3-8B (Zhu et al., 2025) | 41.60 | 9 | 52.43 | 40.87 | 48.94 | 51.05 | 44.77 | 24.95 | 28.63 | 64.20 | 38.62 | 40.96 |
| InternVL3-14B (Zhu et al., 2025) | 45.94 | 5 | 54.32 | 60.17 | 50.35 | 51.81 | 51.45 | 28.04 | 28.26 | 68.04 | 35.37 | 34.46 |
| InternVL3-38B (Zhu et al., 2025) | 48.48 | 2 | 63.42 | 63.58 | 54.59 | 58.29 | 50.55 | 29.90 | 28.52 | 72.16 | 36.76 | 33.49 |
| InternVL3-78B (Zhu et al., 2025) | 49.33 | 1 | 63.78 | 63.12 | 56.24 | 59.24 | 51.45 | 27.63 | 30.19 | 74.51 | 38.46 | 35.90 |
| Qwen-VL2.5-3B (Wang et al., 2024c) | 40.30 | 10 | 55.41 | 47.51 | 46.12 | 42.29 | 44.73 | 32.16 | 23.87 | 59.41 | 33.30 | 30.84 |
| Qwen-VL2.5-7B (Wang et al., 2024c) | 39.18 | 12 | 58.38 | 35.09 | 50.12 | 45.33 | 44.00 | 31.13 | 29.42 | 64.51 | 33.19 | 37.35 |
| Qwen-VL2.5-32B (Wang et al., 2024c) | 47.36 | 4 | 63.06 | 55.09 | 51.76 | 66.29 | 56.91 | 26.39 | 27.48 | 68.04 | 37.50 | 40.24 |
| Qwen-VL2.5-72B (Wang et al., 2024c) | 47.85 | 3 | 58.38 | 60.12 | 50.12 | 59.81 | 53.64 | 26.19 | 33.03 | 71.37 | 36.81 | 36.39 |
| **Specialized Spatial Reasoning Models** | | | | | | | | | | | | |
| SpaceMantis-13B (Chen et al., 2024) | 36.36 | 6 | 47.03 | 36.59 | 40.94 | 34.86 | 33.09 | 22.27 | 24.39 | 49.22 | 38.25 | 39.28 |
| SpaceQwen2.5-VL-3B (Chen et al., 2024) | 40.25 | 3 | 58.11 | 39.88 | 41.18 | 40.95 | 40.91 | 29.90 | 25.81 | 63.73 | 38.83 | 39.76 |
| SpaceThinker-Qwen2.5VL-3B (Chen et al., 2024) | 40.42 | 2 | 47.84 | 53.06 | 43.29 | 35.43 | 38.73 | 24.33 | 28.00 | 58.04 | 35.11 | 31.08 |
| SpatialBot-3B (Cai et al., 2024) | 35.68 | 6 | 43.24 | 38.15 | 32.94 | 29.52 | 41.82 | 28.87 | 22.58 | 47.06 | 36.17 | 37.35 |
| RoboPoint-vicuna-v1.5-7B-lora (Yuan et al., 2024) | 35.85 | 6 | 57.03 | 28.61 | 34.82 | 37.33 | 40.55 | 29.90 | 22.71 | 50.20 | 38.72 | 40.96 |
| RoboPoint-vicuna-v1.5-13B (Yuan et al., 2024) | 34.60 | 5 | 55.68 | 28.15 | 42.82 | 32.19 | 32.55 | 24.12 | 27.74 | 49.02 | 37.66 | 33.49 |
| SoFar-Qwen2.5VL-3B (Qi et al., 2025) | 45.14 | 1 | 56.49 | 51.16 | 54.12 | 53.14 | 52.73 | 31.75 | 22.88 | 71.60 | 36.56 | 41.69 |
| **Human Evaluation** | | | | | | | | | | | | |
| Human | 92.63 | - | 94.62 | 96.07 | 91.38 | 95.11 | 92.15 | 89.02 | 85.90 | 98.53 | 94.30 | 90.26 |

purpose VLMs, reasoning-oriented LLMs, and spatially specialized models, thereby ensuring broad coverage and fair comparison. We consider four groups of state-of-the-art models:

- *Proprietary Models.* These include the GPT-4o and GPT-4.1 families (Hurst et al., 2024), the Claude series (Anthropic, 2024), and Gemini series (Anil et al., 2023). All are accessed through APIs under zero-shot settings with standardized system prompts (see Section D).

- *Reasoning Models.* We categorize models that explicitly employ long chain-of-thought reasoning, often enhanced through reinforcement learning, such as o1, o3, o4-mini (Jaech et al., 2024; OpenAI, 2025b), and Gemini-2.5 (Anil et al., 2023; Reid et al., 2024). Because their outputs are less amenable to strict parsing, we additionally rely on an automatic judge to compare answers against ground truth, following the LLM-as-a-Judge paradigm (Zheng et al., 2023).

- *Open-Source Models.* This set includes Qwen-VL (Bai et al., 2023), InternVL (Zhu et al., 2025), Gemma (Kamath et al., 2025), and LLaVA-OneVision (Liu et al., 2023b). These are locally deployed with standardized prompts to ensure reproducibility.

- *Specialized Spatial Models.* We further benchmark models designed specifically for spatial reasoning, including SpatialVLM (Chen et al., 2024), RoboPoint (Yuan et al., 2024), SpatialBot (Cai et al., 2024), and SoFar (Qi et al., 2025). These systems incorporate explicit spatial signals such as metric 3D information, point affordances, or semantic orientation to improve reasoning.

Table 3: **Camparison of textual Chain-of-Thought and PointGraph on OmniSpatial-test**.

| | | | Dynamic Reasoning | | Spatial Interaction | | | Complex Logic | | Perspective Taking | | |
| --- | --- | --- | --- | --- | --- | --- | --- | --- | --- | --- | --- | --- |
| Method | Avg. | Improve | Manipulate | Motion Analysis | Traffic Analysis | Locate | Geospatial Strategy | Pattern Recognition | Geometric Reasoning | Ego Centric | Allo Centric | Hypothetical |
| **GPT-4.1-mini** | - | - | - | - | - | - | - | - | - | - | - | - |
| (w/o CoT) | 48.86 | - | 64.05 | 58.55 | 57.65 | 59.43 | 56.91 | 28.87 | 34.06 | 68.82 | 37.18 | 41.20 |
| (w/ Zero-shot CoT) | 49.81 | **+0.95** | 62.97 | 58.96 | 59.06 | 62.48 | 58.55 | 27.63 | 32.52 | 69.41 | 40.11 | 40.96 |
| (w/ Manual CoT) | 49.76 | **+0.90** | 65.68 | 58.90 | 58.59 | 64.38 | 56.91 | 28.45 | 32.13 | 69.61 | 39.31 | 41.20 |
| (w/ PointGraph) | 50.49 | **+1.63** | 67.57 | 62.14 | 57.65 | 64.76 | 58.18 | 28.87 | 30.32 | 70.59 | 38.83 | 42.17 |
| **Gemini-2.5-Flash** | - | - | - | - | - | - | - | - | - | - | - | - |
| (w/o CoT) | 51.47 | - | 66.22 | 65.90 | 63.53 | 71.43 | 66.36 | 32.99 | 34.84 | 70.59 | 31.91 | 38.55 |
| (w/ Zero-shot CoT) | 51.53 | **+0.06** | 63.51 | 61.27 | 58.82 | 67.62 | 65.45 | 42.27 | 34.84 | 79.41 | 35.90 | 32.53 |
| (w/ Manual CoT) | 52.12 | **+0.65** | 67.57 | 62.72 | 68.24 | 73.33 | 60.91 | 38.14 | 34.19 | 75.49 | 35.90 | 33.73 |
| (w/ PointGraph) | 53.23 | **+1.76** | 62.16 | 69.94 | 64.71 | 67.62 | 59.09 | 29.90 | 38.06 | 74.51 | 37.77 | 37.35 |
| **Qwen-VL2.5-3B** | - | - | - | - | - | - | - | - | - | - | - | - |
| (w/o CoT) | 41.45 | - | 58.65 | 43.06 | 39.53 | 50.67 | 48.73 | 32.78 | 22.58 | 61.96 | 37.66 | 37.35 |
| (w/ Zero-shot CoT) | 40.64 | **-0.81** | 59.73 | 43.87 | 46.12 | 48.38 | 43.27 | 25.36 | 22.84 | 59.61 | 36.54 | 37.59 |
| (w/ Manual CoT) | 40.07 | **-1.38** | 55.68 | 46.65 | 47.29 | 40.57 | 46.00 | 28.04 | 24.39 | 60.39 | 33.35 | 31.57 |
| (w/ PointGraph) | 44.36 | **+2.91** | 55.68 | 55.20 | 48.94 | 52.19 | 52.36 | 29.90 | 25.55 | 66.08 | 35.11 | 31.08 |

## 4.2 EVALUATION METRICS

We measure accuracy on multiple-choice questions. For standard proprietary and open-source models, we test four output protocols: direct answer, regular-expression parsing, JSON parsing, and LLM-as-a-Judge (Zheng et al., 2023). For reasoning-oriented models with unstructured CoT outputs, correctness is assessed by GPT-4.1-mini against ground truth. Ablations of these evaluation strategies are provided in Section C.1.

## 4.3 MAIN RESULTS

**Overall Model Performance** As illustrated in Table 2, we observe the following findings: **(i)** Proprietary reasoning models, such as ChatGPT o3 (OpenAI, 2025b) and Gemini-2.5-pro (Anil et al., 2023), achieve the highest performance, surpassing a 56% overall success rate; however, there remains a significant gap compared to human-level understanding, and they require a lot of inference time and tokens. **(ii)** Open-source models also demonstrate competitive results, with large-scale models like InternVL3-78B (Zhu et al., 2025) and Qwen-VL2.5-72B (Wang et al., 2024c) achieving comparable performance to GPT-4.1-mini and Gemini-2.0-flash-exp. **(iii)** Specialized Spatial Reasoning Models, due to limitations in dataset coverage and model capacity, struggle to achieve substantial improvements on comprehensive benchmarks.

**Category-wise Analysis** We observe notable performance differences across spatial reasoning categories: **(i)** Leveraging their extensive world knowledge and local understanding capabilities, proprietary models have demonstrated strong performance in Dynamic Reasoning and Spatial Interaction, indicating that reasoning models possess high proficiency in temporal understanding, spatial relationship analysis, and map-based **comprehension. (ii)** For Pattern Geometric Reasoning, which involves spatial imagination in planar geometry, even reasoning models designed for extended thinking can only achieve an accuracy of around 30% to 40%, slightly surpassing the random baseline. **(iii)** Current models exhibit limited perspective-taking abilities, predominantly analyzing scenarios from an ego-centric viewpoint while struggling to imagine perspectives from others' viewpoints.

**Impact of PointGraph & Spatial CoT** To test whether structured segmentation improves performance, we apply PointGraph as a pre-processing step for GPT-4.1, Gemini-2.5-flash and Qwen-VL2.5-7B. Results in Table 3 show a clear accuracy boost, particularly in the Dynamic Reasoning and Perspective-Taking Track, validating the benefits of integrating structured object representation, while traditional textual CoT difficult to bring about significant improvement. Fig. 6 and Table 4 further demonstrates the effectiveness of our proposed Spatial CoT on the OmniSpatial Perspective-Taking track. Through

Table 4: **Performance of Spatial CoT on OmniSpatial Perspective-Taking track.**

| Method | Avg. | Improve | Ego Centric | Allo Centric | Hypothetical |
| --- | --- | --- | --- | --- | --- |
| **GPT-4.1-mini** | – | – | – | – | – |
| (w/ Zero-shot CoT) | 45.56 | - | 69.41 | 40.11 | 40.96 |
| (w/ Spatial CoT) | 47.58 | **+2.02** | 69.43 | 42.37 | 44.34 |
| **Qwen-VL2.5-3B** | – | – | – | – | – |
| (w/ Zero-shot CoT) | 40.89 | - | 59.61 | 36.54 | 37.59 |
| (w/ Spatial CoT) | 42.90 | **+2.01** | 60.80 | 39.25 | 37.44 |

Table 5: Training Exploration on OmniSpatial-test.

| Method | Avg. | Improve | Manipulate | Motion Analysis | Traffic Analysis | Locate | Geospatial Strategy | Pattern Recognition | Geometric Reasoning | Ego Centric | Allo Centric | Hypothetical |
|---|---|---|---|---|---|---|---|---|---|---|---|---|
| **Qwen-VL2.5-3B** | - | - | - | - | - | - | - | - | - | - | - | - |
| Zero-shot | 40.30 | - | 55.41 | 47.51 | 46.12 | 42.29 | 44.73 | 32.16 | 23.87 | 59.41 | 33.30 | 30.84 |
| + OmniSpatial-train (6.9K) | **48.12** | **+7.82** | **59.19** | **51.16** | **50.12** | **51.81** | **45.45** | **34.02** | 31.74 | 68.63 | **36.76** | **35.90** |
| + Template corpus (200K) | 41.59 | +1.29 | 50.90 | 46.71 | 40.94 | 45.33 | 40.55 | 28.87 | 24.39 | **72.55** | 35.11 | 33.01 |

Table 6: Generalization on VSI-Bench. Training with OmniSpatial yields consistent gains.

| Method | overall | Appearance Order | Obj Abs Distance | Obj Counting | Obj Rel Distance | Obj Size | Room Size | Route Planning | Obj Rel Direction |
|---|---|---|---|---|---|---|---|---|---|
| **Qwen-VL2.5-3B** | - | - | - | - | - | - | - | - | - |
| Zero-shot | 34.06 | 34.63 | 11.39 | 40.18 | **36.20** | 46.11 | 38.58 | 31.96 | 36.92 |
| + SpaceR-7B | 41.68 | 46.60 | **19.63** | 55.27 | 34.23 | 58.32 | 35.17 | **34.54** | 42.84 |
| + OmniSpatial | **43.68** | **58.25** | 15.13 | **57.36** | 34.37 | **60.99** | **41.11** | 34.02 | **44.16** |

novel view synthesis facilitated by InstantMesh, both GPT-4.1 and Qwen-VL2.5-7B exhibit significant performance improvements, validating the effectiveness of explicit spatial imagination.

## 4.4 TRAINING EXPLORATION

**Supervised Training on OmniSpatial-train.** We further study whether OmniSpatial-train can effectively teach spatial skills instead of merely fitting templates. Starting from a strong open-source baseline, supervised fine-tuning on 6.9K samples yields a substantial +7.82 point average gain over zero-shot, with consistent improvements across dynamic, interaction, and perspective-taking oriented tracks. In contrast, training on a much larger 200K template-style corpus that follows the construction process of VSI-Bench (Yang et al., 2024) brings only a marginal +1.29 average gain, underscoring the value of diverse, manually curated spatial tasks over synthetic templates.

**Generalization to VSI-Bench.** To examine cross-benchmark generalization, we adopt the SpaceR-7B pipeline and compare supervised training with and without OmniSpatial. Adding OmniSpatial improves the overall score on *VSI-Bench* from 41.68 to 43.68. Notably, it boosts categories requiring ordering, counting, and metric/room-size reasoning (e.g., *appearance_order*, *obj_counting*, *obj/room_size*). These results indicate that OmniSpatial provides complementary supervision that transfers to external spatial tasks rather than overfitting to in-benchmark patterns.

## 5 RELATED WORKS

### 5.1 BENCHMARKING SPATIAL REASONING

Various studies have introduced innovative benchmarking methodologies (Szymanska et al., 2024; Chen et al., 2024; Cheng et al., 2024; Cai et al., 2024; Song et al., 2024; Qi et al., 2025; Ray et al., 2024) to advance spatial reasoning evaluation. Spatial VQA (Du et al., 2024) was among the first to incorporate spatial information into vision-language models, enabling fundamental spatial relationship reasoning. SpatialBot (Cai et al., 2024) categorized spatial reasoning into various hierarchical levels, extending its applicability to robotic manipulation tasks. RoboSpatial (Song et al., 2024) proposes a large-scale template-based spatial relationship dataset, focusing on positional relationships from different perspectives. VSI-Bench (Yang et al., 2024) combined video data (Chandrasegaran et al., 2024; Liu et al., 2023c; Fu et al., 2024a; Li et al., 2024b; Fang et al., 2024b) with cognitive maps (Momennejad et al., 2023; Apostolopoulos & Groumpos, 2023) to simulate human-like spatial cognition and optimize reasoning in dynamic environments. Recently, SoFar (Qi et al., 2025) introduced the 6-DoF SpatialBench to evaluate the understanding of orientation.

While these benchmarks have made significant contributions, a unified framework encompassing a wide range of complex spatial reasoning tasks remains lacking. Inspired by prior spatial reasoning research (Xu et al., 2024b; Wang et al., 2024a; Lin et al., 2014; Nwankwo et al., 2024; Wang et al., 2024b), we identified several limitations in existing benchmarks, such as reliance on generated images (Szymanska et al., 2024; Fu et al., 2024b; Kamath et al., 2023; Liu et al., 2023a; Rajabi & Kosecka, 2023; Shiri et al., 2024), LLM-generated templates (Linghu et al., 2024; Du et al., 2024; Rädsch et al., 2025), and domain-specific focus (Xie et al., 2025; Chow et al., 2025; Danish et al.,

2024). These issues hinder their comprehensiveness and real-world applicability. To address these gaps, our study proposes a comprehensive and integrative spatial reasoning benchmark.

## 5.2 Spatial Vision-Language Models

Spatial vision-language models integrate computer vision (He et al., 2022; Oquab et al., 2024; Radford et al., 2021) and natural language processing (Bai et al., 2023; Wang et al., 2024c; Brown et al., 2020; Anil et al., 2023; Touvron et al., 2023a;b; Zhang et al., 2024b) to enhance machine understanding of spatial relationships and mental intelligence. Recent research (Qi et al., 2024; Chen et al., 2024; Cheng et al., 2024; Cai et al., 2024; Song et al., 2024; Yuan et al., 2024; Qi et al., 2025; Chen et al., 2025b) has increasingly focused on extending vision-language models to support spatial reasoning in dynamic and 3D environments. One of the pioneering works in this domain, SpatialVLM (Chen et al., 2024), has significantly advanced spatial reasoning by constructing an RGBD-based visual question-answering dataset. GS-Reasoner Chen et al. (2025b) introduces 3D grounding as a chain-of-thoughts for spatial reasoning tasks. These models (Hurst et al., 2024; Reid et al., 2024; Wang et al., 2024c) effectively process multimodal data containing spatial information, serving as a bridge between visual perception and linguistic reasoning.

Further advancements (Liu et al., 2024b;a; Ramakrishnan et al., 2024; Tang et al., 2024; Yang et al., 2023; Rozanova et al., 2021; Wu et al., 2024; Stogiannidis et al., 2025) include SpatialRGPT (Cheng et al., 2024), which extends RGB-D spatial understanding by incorporating 3D scene graphs and spatial data to improve inference capabilities. Similarly, SpatialBot (Cai et al., 2024) explores hierarchical deep reasoning mechanisms to handle depth and spatial structures in complex environments. Recently, SoFar (Qi et al., 2025) proposed semantic orientation and trained an open-world orientation model to enhance the orientation understanding of vision-language models, significantly improving spatial understanding and robotic operation capabilities (Cho et al., 2024; Driess et al., 2023; Mu et al., 2023; Huang et al., 2024b; Nasiriany et al., 2024; Huang et al., 2024a; Qi et al., 2025; He et al., 2025a; Zhang et al., 2025b). However, existing research remains limited in addressing the full complexity and comprehensiveness of spatial reasoning tasks. To bridge this gap, our study aims to develop a more comprehensive benchmark that rigorously evaluates spatial reasoning capabilities.

## 6 Conclusion

We introduce OmniSpatial, a benchmark for comprehensive visual–spatial reasoning. OmniSpatial distills spatial cognition into four primary categories—dynamic reasoning, complex logic, spatial interaction, and perspective-taking—spanning 50 fine-grained subtasks and 8.4 K manually-curated question–answer pairs. Extensive experiments show that state-of-the-art proprietary and open-source VLMs peak at 57% accuracy—over 30 points below human performance—struggling especially with geometric reasoning and non-egocentric perspective taking. To bridge these gaps, we introduce **PointGraph** for structured scene-graph reasoning and **SpatialCoT** for viewpoint-aware CoT, both yielding consistent gains and underscoring the value of structured and multi-view reasoning.

## Ethic Statement

This work complies with the ICLR Code of Ethics. All datasets are publicly available and used under their respective licenses. The research raises no direct ethical or legal concerns, and the authors are committed to responsible and fair use of the proposed methods.

## Reproducibility Statement

We have made every effort to ensure the reproducibility of our work. The proposed model, implementation details, and evaluation protocols are described in detail in the main paper and appendix. All datasets used are publicly available and properly referenced. To further support reproducibility, we release the source code in the supplementary materials.

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

# Contents

## A   DETAILED TASK DESIGN

OmniSpatial aims to comprehensively evaluate the spatial reasoning capabilities of Vision-Language Models, covering four major categories: Dynamic Reasoning, Complex Logic, Spatial Interaction, and Perspective Taking. Each category not only focuses on different types of spatial reasoning tasks but also includes challenges based on real-world application scenarios. This approach helps researchers better understand and enhance models' multi-domain spatial reasoning abilities. The following presents the underlying considerations and practical value behind each task design.

### A.1   DYNAMIC REASONING

The Dynamic Reasoning category focuses on the model's understanding of object movement and its changes, assessing the ability to make accurate judgments in uncertain or rapidly changing environments. Spatial dynamics are critical not only in robot control but also have broad applications in fields like autonomous driving and intelligent surveillance.

#### A.1.1   MANIPULATION

**Operational Position Selection** This task evaluates how models determine the optimal interaction point with objects in complex environments. Selecting the best grasping point can prevent tilting or damage to objects and improve the efficiency and precision of operations. This task is crucial in robotic grasping, especially when environmental conditions are unstable.

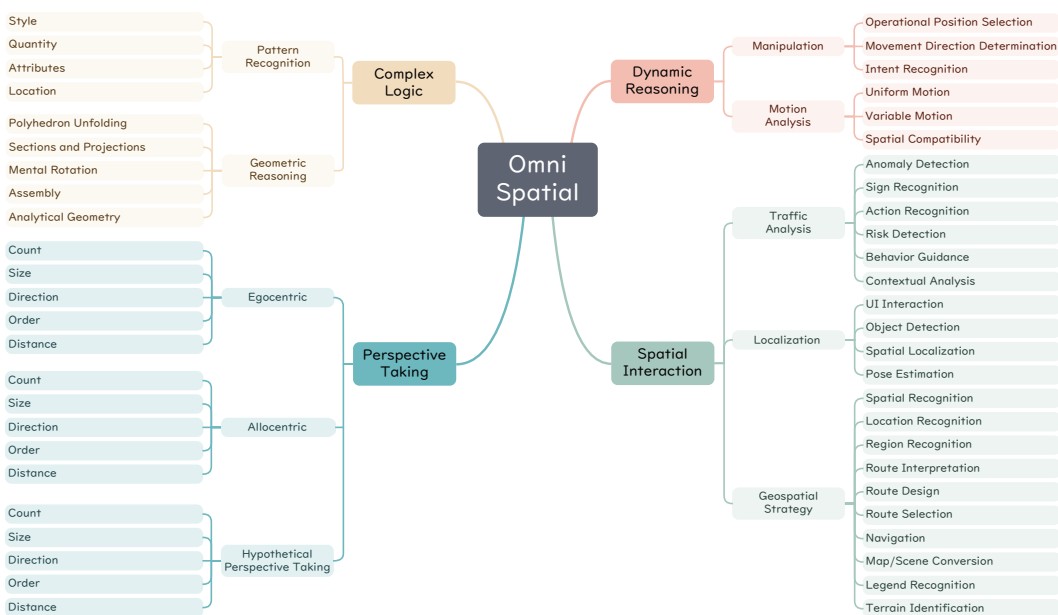

Figure 7: **OmniSpatial tasks**.The tasks are organized into three levels, with each of the four categories of spatial abilities containing no fewer than two subtasks. The final level features a more detailed subdivision, inspired by real-life scenarios
.

**Movement Direction Determination** This task assesses the model's ability to predict the movement direction of an object or itself, providing decision support for automated systems and helping to optimize robot motion strategies.

**Intent Recognition** Intent recognition involves inferring the purpose or goal behind a movement. This task is particularly important for contextual analysis, such as determining whether a person is reaching for a door handle to open or close a door. This capability enhances the reasoning of human-robot interactions and optimizes the interaction experience of intelligent assistants and robots.

### A.1.2 MOTION ANALYSIS

**Uniform Motion** The model's ability to reason about uniform motion reflects its fundamental understanding of time and spatial relationships, such as estimating the speed or time required for a target to move. This is applicable in areas like object tracking and path prediction, such as estimating vehicle travel time or train arrival time.

**Variable Motion** Variable motion analysis focuses on understanding acceleration and deceleration processes. By predicting the position changes of an object during variable motion, the model can better simulate dynamic phenomena in the physical world, such as calculating braking distance for vehicles. This is critical in autonomous driving and robot control.

**Spatial Compatibility** Tasks that assess whether an object fits within a specific space directly relate to the precision of robots and automated devices in real-world operations. For instance, determining whether luggage can fit into an overhead compartment is applicable in logistics and automated warehouses, helping systems make effective object adaptation decisions.

### A.2 COMPLEX LOGIC

The Complex Logic category focuses on higher-order spatial reasoning, including tasks such as geometric transformations and pattern recognition, challenging the model's ability to abstractly understand and reason in multi-dimensional spatial environments.

### A.2.1 PATTERN RECOGNITION

**Style** Style recognition tasks assess the model's ability to infer visual rules in structured patterns. Typical operations include completing missing parts, combining or subtracting shapes, comparing similarities and differences, and performing visual logic like black-white inversion. This skill is crucial for tasks like intelligence tests and diagrammatic reasoning.

**Quantity** Quantity-based tasks evaluate the model's ability to perform visual numerosity reasoning, focusing on implicit quantitative patterns such as the number of points, lines, regions, elements, or strokes. These tasks require abstract counting under diverse spatial configurations, often without explicit numerical labels or symbols.

**Attributes** Attribute-based tasks evaluate the model's ability to reason about non-numeric visual properties such as symmetry (axial or radial), curvature, and openness. These tasks require the recognition of structural features that do not rely on quantity but rather on geometric or perceptual traits.

**Location** Location tasks evaluate the model's ability to reason about spatial changes such as translation, rotation, and reflection. The focus is on how objects shift in space while preserving their structure. This skill is essential for grid-based reasoning, motion prediction, and geometric pattern understanding.

### A.2.2 GEOMETRIC REASONING

**Polyhedron Unfolding** This task examines whether the model can infer the 2D net of a 3D object, reflecting its ability to mentally construct spatial layouts. It is particularly relevant to applications such as packaging design, industrial manufacturing, and aerospace engineering.

**Sections and Projections** By evaluating how models interpret 2D cross-sections or projections from different viewpoints, this task challenges their capacity to connect visual appearance with internal structure—key to fields like medical imaging, architecture, and mechanical design.

**Mental Rotation** Mental rotation requires the model to simulate the rotation of objects in mind and track changes across views. It underpins spatial reasoning in tasks ranging from CAD modeling to object manipulation in virtual and augmented reality.

**Assembly** This task involves reasoning about how separate parts fit together into a coherent whole, testing the model's understanding of geometric constraints. It has broad relevance in robotics, structural analysis, and physical assembly planning.

**Analytical Geometry** These tasks are inspired by classic geometry problems, requiring models to reason about spatial relations using angles, distances, and symmetry. They bridge mathematical logic with visual structure, supporting applications in structured reasoning and spatial abstraction.

## A.3   SPATIAL INTERACTION

The Spatial Interaction category evaluates a model's ability to reason about interactions with objects and environments. It includes tasks such as *Traffic Analysis*, *Localization*, and *Geospatial Strategy*, reflecting the model's understanding and application of spatial knowledge in real-world scenarios.

### A.3.1   TRAFFIC ANALYSIS

**Anomaly Detection** This task focuses on identifying potential dangers or traffic violations in complex scenes, such as unsafe following distances or unusual vehicle behavior. It plays a key role in ensuring safety in autonomous driving systems.

**Sign Recognition** This task evaluates the model's ability to detect and interpret traffic signs, including speed limits, no-entry zones, and yield signs. Accurate recognition is critical for safe and rule-compliant decision-making.

**Action Recognition** This task involves identifying or predicting the actions of traffic participants, such as driver gestures, police signals, or pedestrian intentions. It is important for understanding dynamic human behavior in traffic environments.

**Risk Detection** This task aims to detect immediate hazards in the environment, such as an opening car door or a pedestrian crossing the road. Timely detection supports effective avoidance and control strategies.

**Behavior Guidance** This task provides context-aware behavioral suggestions, such as advising to turn off high beams or reminding that parking is prohibited. It enhances overall driving safety and compliance.

**Contextual Analysis** This task assesses the model's ability to interpret spatial relations and behaviors based on environmental cues. For example, estimating wind conditions to anticipate overtaking behavior or understanding road status to infer potential risks.

### A.3.2   LOCALIZATION

**UI Interaction** This task requires the model to determine which icon should be selected within a user interface based on contextual understanding, and to accurately localize its position. It reflects the model's ability to integrate semantic interpretation with spatial reasoning, supporting applications in intelligent assistants and automated interface control.

**Object Detection** This task involves identifying specific target objects within an image. It is often paired with spatial localization to jointly assess what the object is and where it is located.

**Spatial Localization** This task focuses on determining the precise position of objects within a scene. It is commonly evaluated alongside object detection to answer questions like "What is at this location?" or "Where is this object?"

**Pose Estimation**  This task estimates the orientation and spatial configuration of objects, such as detecting whether a cup is upright or tipped over. It is frequently integrated with spatial localization to enable more nuanced scene understanding.

### A.3.3  GEOSPATIAL STRATEGY

**Spatial Recognition**  Assesses the model's capacity to identify spatial structures such as rooms, corridors, or zones within a scene. This is essential for semantic navigation and indoor mapping.

**Location Recognition**  This task evaluates the model's ability to identify specific locations on a map or scene, such as recognizing the position marked as "You are here" or locating a designated landmark. It reflects the model's capacity to associate spatial markers with real-world positions.

**Region Recognition**  Focuses on distinguishing and classifying regions in a broader spatial context, such as residential vs. industrial zones on a map.

**Route Interpretation**  Tests the model's ability to follow or explain a route depicted in a map or scene. It requires understanding directional arrows, route labels, and spatial transitions.

**Route Design**  Involves selecting or generating an optimal path to reach a given goal, considering spatial constraints and possible alternatives.

**Route Selection**  Compares multiple candidate routes and chooses the most suitable one based on efficiency, safety, or contextual requirements.

**Navigation**  Evaluates the model's ability to understand *smartphone or in-vehicle navigation interfaces*, including interpreting turn-by-turn directions, identifying route segments, and understanding map overlays. This is crucial for building intelligent voice assistants and real-time guidance systems.

**Map/Scene Conversion**  Tests the ability to mentally convert between map views and real-world scenes, which is critical in correlating schematic representations with physical surroundings.

**Legend Recognition**  It requires identifying and interpreting map symbols (e.g., stairs, elevators, emergency exits) and a foundational skill in navigation and spatial reasoning.

**Terrain Identification**  Focuses on distinguishing types of terrain (e.g., flat, uphill, water-crossing), which is essential for planning safe and feasible paths in outdoor navigation or robotics.

### A.4  PERSPECTIVE TAKING

This category evaluates the model's ability to understand spatial relationships from different viewpoints. Since changes in perspective directly affect what is observed, the ability to reason across varying angles is essential for robotic perception and interaction.

### A.4.1  EGOCENTRIC

**Count**  Counting the number of visible objects from the current perspective is crucial for dynamic interaction. For example, in robotic grasping tasks, knowing how many targets are visible helps determine the appropriate operation strategy.

**Size**  Judging the size of an object from the observer's viewpoint aids robots or virtual systems in depth perception, helping assess whether an object can be grasped or properly placed.

**Direction**  This refers to the direction directly seen by the observer. It is especially important in autonomous driving scenarios, where understanding object movement helps predict traffic conditions and enables timely responses.

**Order**  Analyzing the arrangement of multiple objects in an image is essential for robotic operations, helping prioritize which objects to interact with first.

**Distance**  The distance between objects as perceived from the observer's viewpoint is a key capability in navigation systems, enabling path planning and obstacle avoidance.

### A.4.2 ALLOCENTRIC

**Count** Understanding how the perceived quantity of objects changes under different viewpoints is key. Due to variations in position and orientation, occlusion often occurs—for example, a driver may have blind spots, while a road surveillance camera can see objects hidden in those areas. This task evaluates the model's ability to judge the difference in object counts from various observation points.

**Size** This task involves evaluating an object's size from a specified observer's viewpoint. Due to the general principle that closer objects appear larger and distant ones appear smaller, objects of the same size may look different to different observers. The model is expected to either infer the true size based on reference objects or estimate how perceived size changes with viewing position.

**Direction** This task emphasizes judging directions from abstract positions, such as another agent's viewpoint or a map. The answers often differ from what is directly observed, requiring one to adopt the target's perspective—engaging in perspective-taking. It is crucial not only for large-scale path planning but also for understanding an object's intrinsic orientation.

**Order** This task requires observing the arrangement of objects from a specified viewpoint—for example, the seating order of students in the front row as seen by a teacher on the podium, which is exactly reversed from what a camera at the back of the classroom would capture. Only by understanding "what the target sees" can one make accurate predictions or judgments about the scene.

**Distance** Differences in the observer's position and orientation lead to variations in perceived size and distance. For example, in the same scene, a photographer taking pictures of the same object from different camera angles will capture different impressions of its size and distance. Changing the camera position essentially means changing the perspective. This ability is vital for coordination and environmental assessment in human-robot interaction or multi-robot collaborative tasks.

### A.4.3 HYPOTHETICAL PERSPECTIVE TAKING

This category focuses on imagining the scene from a specified but non-existent viewpoint, requiring the model to mentally adopt a fictional position—an advanced form of perspective-taking in spatial reasoning.

**Count** Predicting how many target objects would be visible from a hypothetical viewpoint—for example, a person standing at the opposite corner of the street may see a different distribution of objects. The model must reason about occlusion, orientation, and visibility from the imagined perspective.

**Size** Inferring the apparent size of objects from a hypothetical location and direction. For instance, the same object may appear larger when viewed up close or smaller when viewed from above. The model needs to simulate how visual scale changes with altered viewpoints.

**Direction** Reasoning about how an object's direction appears from a location where no observer is present. For example, a pedestrian walking toward a doorway would appear "head-on" from the entrance, but present a different direction from a side view.

**Order** Simulating the arrangement of objects from another location helps assess how spatial sequences change across viewpoints. For example, the seating order seen from the podium may be the reverse of what's seen from the back of the room.

**Distance** Estimating relative distances between objects from a hypothetical position requires mentally adopting a new viewpoint. This supports effective planning and coordination in tasks such as multi-robot navigation or collaborative manipulation.

## B ADDITIONAL RELATED WORKS

### B.1 SPATIAL REASONING IN PSYCHOLOGY

In psychology, spatial reasoning refers to an individual's ability to acquire, organize, utilize, and adapt spatial knowledge, recognized as one of the nine primary reasonings (Gardner, 2011). To

Table 7: **Comparative analysis of various prompting and evaluation strategies**.

| Prompt Type | Eval Type | Avg. | Dynamic Reasoning | | Spatial Interaction | | | Complex Logic | | Perspective Taking | | |
| | | | Manipulate | Motion Analysis | Traffic Analysis | Locate | Geospatial Strategy | Pattern Recognition | Geometric Reasoning | Ego Centric | Allo Centric | Hypothetical |
|---|---|---|---|---|---|---|---|---|---|---|---|---|---|
| **GPT-4.1-mini** | - | - | - | - | - | - | - | - | - | - | - | - |
| None | Direct | 39.22 | 57.57 | 36.18 | 58.12 | 44.95 | 53.45 | 12.58 | 18.58 | 62.94 | 37.61 | 37.83 |
| None | RE | 48.86 | 64.05 | 58.55 | 57.65 | 59.43 | 56.91 | 28.87 | 34.06 | 68.82 | 37.18 | 41.20 |
| Zero-shot CoT | RE | 49.81 | 62.97 | 58.96 | 59.06 | 62.48 | 58.55 | 27.63 | 32.52 | 69.41 | 40.11 | 40.96 |
| Manual CoT | RE | 49.76 | 65.68 | 58.90 | 58.59 | 64.38 | 56.91 | 28.45 | 32.13 | 69.61 | 39.31 | 41.20 |
| None | JSON | 48.23 | 61.08 | 55.55 | 57.41 | 58.86 | 54.00 | 28.89 | 29.16 | 70.20 | 40.96 | 40.48 |
| Zero-shot CoT | JSON | 48.70 | 58.67 | 57.17 | 57.88 | 57.71 | 55.09 | 28.89 | 29.29 | 69.02 | 40.16 | 42.89 |
| Manual CoT | JSON | 48.87 | 64.32 | 56.53 | 59.06 | 60.19 | 56.36 | 29.28 | 30.19 | 72.55 | 39.57 | 39.28 |
| None | LLM | 48.02 | 60.54 | 56.82 | 58.59 | 58.10 | 57.45 | 28.04 | 32.90 | 68.24 | 37.55 | 38.31 |
| Zero-shot CoT | LLM | 48.36 | 64.05 | 56.71 | 58.59 | 58.86 | 57.27 | 27.84 | 33.29 | 67.06 | 38.30 | 38.80 |
| Manual CoT | LLM | 49.85 | 62.97 | 59.48 | 58.12 | 61.33 | 58.36 | 28.04 | 31.61 | 69.02 | 41.12 | 39.28 |
| **Gemini-2.5-flash** | - | - | - | - | - | - | - | - | - | - | - | - |
| None | LLM | 51.47 | 66.22 | 65.90 | 63.53 | 71.43 | 66.36 | 32.99 | 34.84 | 70.59 | 31.91 | 38.55 |
| Zero-shot CoT | LLM | 51.53 | 63.51 | 61.27 | 58.82 | 67.62 | 65.45 | 42.27 | 34.84 | 79.41 | 35.90 | 32.53 |
| Manual CoT | LLM | 52.12 | 67.57 | 62.72 | 68.24 | 73.33 | 60.91 | 38.14 | 34.19 | 75.49 | 35.90 | 33.73 |
| **Qwen-VL2.5-3B** | - | - | - | - | - | - | - | - | - | - | - | - |
| None | Direct | 44.04 | 60.27 | 52.20 | 47.53 | 50.48 | 52.73 | 22.68 | 30.32 | 65.49 | 36.49 | 30.84 |
| None | RE | 41.45 | 58.65 | 43.06 | 39.53 | 50.67 | 48.73 | 32.78 | 22.58 | 61.96 | 37.66 | 37.35 |
| Zero-shot CoT | RE | 40.64 | 59.73 | 43.87 | 46.12 | 48.38 | 43.27 | 25.36 | 22.84 | 59.61 | 36.54 | 37.59 |
| Manual CoT | RE | 40.07 | 55.68 | 46.65 | 47.29 | 40.57 | 46.00 | 28.04 | 24.39 | 60.39 | 33.35 | 31.57 |
| None | JSON | 38.08 | 62.97 | 32.49 | 46.59 | 48.95 | 47.64 | 24.54 | 23.61 | 62.16 | 35.85 | 27.47 |
| Zero-shot CoT | JSON | 39.20 | 61.35 | 40.40 | 48.94 | 45.52 | 47.27 | 22.27 | 24.00 | 65.29 | 33.51 | 27.71 |
| Manual CoT | JSON | 38.37 | 58.11 | 34.28 | 42.35 | 46.29 | 48.55 | 27.63 | 24.65 | 62.75 | 36.54 | 26.75 |
| None | LLM | 42.10 | 66.22 | 43.99 | 44.00 | 51.24 | 50.73 | 27.84 | 25.42 | 70.00 | 35.90 | 29.40 |
| Zero-shot CoT | LLM | 35.89 | 54.59 | 36.24 | 49.41 | 40.19 | 40.18 | 24.95 | 20.13 | 61.37 | 29.20 | 33.98 |
| Manual CoT | LLM | 36.26 | 53.78 | 34.10 | 50.35 | 42.10 | 44.00 | 23.51 | 26.06 | 60.39 | 31.38 | 23.86 |

systematically characterize this ability, Buckley et al. (2018)proposed a factor analysis-based framework, distinguishing between spatial visualization, spatial relations (such as mental rotation), and spatial orientation. Malanchini et al. (2020) further identified a strong correlation between spatial orientation and object manipulation skills. Additionally, Newcombe & Shipley (2014)introduced a classification system for spatial thinking, dividing spatial reasoning into two dimensions: intrinsic-extrinsic and static-dynamic. Furthermore, many studies also have contributed to the development of frameworks (Wai et al., 2009; Lee & Bednarz, 2012; Malanchini et al., 2020; Hegarty, 2010; McGarvey et al., 2018) and evaluation methods (Eliot & Smith, 1983; Uttal et al., 2024; 2013; Hegarty & Waller, 2005) for spatial reasoning. Their framework highlights the distinction between object-centric spatial properties and external reference frames, offering valuable insights for applications such as navigation and path planning. These psychological frameworks provide complementary perspectives for understanding and assessing spatial reasoning, offering theoretical foundations for embodied intelligent systems. Inspired by these classifications, our study proposes a novel taxonomy for visual-spatial reasoning, aiming to advance spatial reasoning research in vision-language models.

## C ADDITIONAL ABLATION STUDIES

### C.1 EVALUATION AND FORMAT STRATEGY

The choice of evaluation format and answer-extraction strategy can materially influence how fully a large language model's capabilities are expressed. Because our benchmark introduces a novel visual–spatial reasoning task, Table 7 reports the performance impact of different prompting and evaluation schemes. We consider three prompting paradigms—Direct QA, Zero-shot CoT, and Manual CoT—and three evaluation methods: (i) regex-based field extraction (RE), in which the VLM is instructed to place its answer in a fixed slot at the end of the response; (ii) JSON extraction, where the answer must appear in a dedicated answer field; and (iii) LLM-based free-form evaluation, for which we employ GPT-4.1-mini as an adjudicator.

We observe that reasoning-centric models that strongly obey formatting directives (e.g., Gemini-2.5-flash) cannot be reliably scored with regex extraction, because their compulsory chain-of-thought interferes with the required output template. For standard models, all three evaluation strategies yield comparable scores, and both CoT variants consistently outperform the Direct QA baseline by a small margin. Accordingly, in the main results table, we evaluate non-reasoning models with the

Table 8: **Evaluation on the full OmniSpatial benchmark.** Dark green marks the best *Avg.* accuracy and light green marks the second-best *Avg.* accuracy in all the models.

| Method | Avg. | Rank | Dynamic Reasoning | | | Spatial Interaction | | Complex Logic | | Perspective Taking | | |
| | | | Manipulate | Motion Analysis | Traffic Analysis | Locate | Geospatial Strategy | Pattern Recognition | Geometric Reasoning | Ego Centric | Allo Centric | Hypothetical |
|---|---|---|---|---|---|---|---|---|---|---|---|---|
| **Proprietary Models** | | | | | | | | | | | | |
| GPT-4o | 44.47 | 5 | 60.00 | 57.97 | 62.38 | 55.24 | 59.09 | 29.90 | 31.01 | 42.06 | 24.34 | 28.05 |
| GPT-4.1 | 48.75 | 4 | 61.18 | 66.67 | 66.60 | 69.52 | 60.91 | 30.93 | 35.58 | 47.06 | 27.26 | 33.06 |
| o3-04-16 | 54.52 | 2 | **71.05** | 63.64 | **70.61** | 69.05 | 61.70 | **42.55** | 47.15 | **53.35** | 30.98 | **46.28** |
| Gemini-2.0-Flash | 42.41 | 7 | 56.47 | 56.76 | 55.95 | 63.81 | 57.27 | 15.46 | 30.04 | 43.20 | 24.56 | 28.05 |
| Gemini-2.5-Flash | 51.80 | 3 | 54.84 | **70.43** | 65.33 | 66.00 | 68.75 | 39.53 | 39.46 | 52.70 | 33.49 | 36.84 |
| Gemini-2.5-Pro | **55.05** | **1** | 66.67 | 68.34 | 69.10 | **78.85** | **71.15** | 36.17 | **50.17** | 50.65 | **39.57** | 37.89 |
| **Open-source Models** | | | | | | | | | | | | |
| Qwen-VL-2.5-3B | 39.76 | 8 | 57.65 | 47.34 | 49.07 | 49.52 | 58.18 | 29.90 | 26.97 | 40.62 | 30.47 | 29.88 |
| Qwen-VL-2.5-32B | 43.97 | 6 | 58.82 | 52.42 | 52.97 | 68.57 | 51.82 | 27.84 | 29.51 | 51.70 | 26.82 | 35.98 |

Manual CoT + RE setting, whereas reasoning-oriented models are assessed with Manual CoT + LLM evaluation.

## C.2 Evaluation on the full OmniSpatial benchmark

To complement the main experiments, we further conduct evaluation on the entire 8.4K OmniSpatial test set without applying the train/test split. This setting measures model performance on all available annotated questions, thereby reducing variance due to subset sampling and providing a more holistic view of spatial reasoning ability. Table 8 reports the results across all twelve fine-grained tracks.

Overall, the relative ranking of models remains stable compared to the smaller-scale splits discussed in the main text. Gemini-2.5-Pro achieves the highest average score of 55.05, securing Rank 1 among all compared systems. ChatGPT o3 follows closely with 54.52, while Gemini-2.5-Flash achieves 51.80. Other proprietary systems, such as GPT-4.1 and GPT-4o, also maintain consistent performance. On the open-source side, Qwen-VL-2.5-32B achieves the strongest result with 43.97, outperforming other Qwen variants and aligning with the trends observed in the evaluations.

A closer per-track analysis highlights complementary strengths across models. ChatGPT o3 excels at *Manipulate*, *Traffic Analysis*, and also leads in *Pattern Recognition*, *Ego-Centric*, and *Hypothetical* reasoning. In contrast, Gemini-2.5-Pro dominates *Locate*, *Geospatial Strategy*, *Geometric Reasoning*, and *Allo-Centric* perspective taking. Gemini-2.5-Flash yields the highest accuracy on *Motion Analysis*. These complementary strengths suggest that different architectures capture distinct aspects of spatial reasoning, and point towards potential benefits of ensembling or multi-expert distillation.

## C.3 Details of Human Baseline & Inter-Annotator Agreement

To establish an upper bound of performance, we conducted a human evaluation on the OmniSpatial benchmark. Six human annotators (graduate students with backgrounds in vision and robotics) were recruited. Each participant was presented with a randomized subset of questions covering all four tracks and a balanced selection of fine-grained tasks. The interface was blinded: multiple-choice options were randomized and no additional context was provided. Annotators were instructed to answer independently without external resources.

Each item was labeled by three different annotators, and the final answer was obtained via majority vote. To quantify annotation reliability, we report inter-annotator agreement (IAA) using Krippendorff's $\alpha$ for multi-annotator agreement. Table 9 summarizes the results across tracks. We also report 95% confidence intervals (bootstrap over questions) for human accuracy.

These results indicate that human annotators achieve $\sim$89% average accuracy, with substantial agreement across annotators ($\kappa/\alpha \approx 0.84$). Even the most abstract and complex forms of spatial reasoning achieve a consistency score of 0.76. This provides a reliable estimate of the human upper bound and confirms the internal consistency of the benchmark.

Table 9: Human baseline accuracy and inter-annotator agreement (IAA) across tracks.

| Track | Accuracy (%) | Krippendorff's $\alpha$ |
|---|---|---|
| Dynamic Reasoning | $95.2 \pm 1.3$ | 0.92 |
| Spatial Interaction | $93.5 \pm 1.6$ | 0.85 |
| Complex Logic | $87.9 \pm 5.5$ | 0.76 |
| Perspective Taking | $94.4 \pm 2.2$ | 0.80 |
| **Overall** | $\mathbf{92.6 \pm 2.5}$ | **0.84** |

## D  SYSTEM PROMPTS

We present all the system prompts used in our experiments in Figs. 13 and 14 to facilitate reproducibility. We observe that some models are sensitive to the choice of system prompt, which may stem from distributional biases in their training data. In contrast, inference-oriented models generally exhibit stronger generalization capabilities and are less reliant on carefully crafted prompts. We conduct extensive experiments and iterative tuning on the system prompts, with the ultimate goal of objectively and faithfully evaluating each model's actual capability without being confounded by prompt-induced bias.

## E  ADDITIONAL VISUALIZATION

We present more examples of question-answer pairs from the OmniSpatial, with perspective taking, spatial interaction, dynamic reasoning, and complex logic, shown in Figs. 15 to 18, respectively. Our benchmark comprises a rich collection of data samples spanning diverse scenarios, resolutions, lighting conditions, and geographical regions. It includes both absolute numerical analyses and relative spatial relationships, aiming to comprehensively evaluate the spatial reasoning capabilities of vision-language models. To the best of our knowledge, our spatial intelligence benchmark is the most diverse and comprehensive to date, and is entirely human-annotated, enabling a faithful evaluation of models' visual–spatial reasoning capabilities without the confounding effects of templated patterns.

## F  ADDITIONAL EXPERIMENTS

### F.1  THE SYNERGY OF POINTGRAPH AND SPATIALCOT

Table 10: **Performance of Spatial CoT on OmniSpatial Perspective-Taking track.**

| Method | Avg. | Improve | Ego Centric | Allo Centric | Hypothetical |
|---|---|---|---|---|---|
| **GPT-4.1-mini** | – | – | – | – | – |
| (w/ Zero-shot CoT) | 45.56 | - | 69.41 | 40.11 | 40.96 |
| (w/ Spatial CoT) | 47.58 | **+2.02** | 69.43 | 42.37 | 44.34 |
| (w/ Spatial CoT & PointGraph) | 48.70 | **+3.14** | 71.09 | 42.52 | 44.77 |
| **Qwen-VL2.5-3B** | – | – | – | – | – |
| (w/ Zero-shot CoT) | 40.89 | - | 59.61 | 36.54 | 37.59 |
| (w/ Spatial CoT) | 42.90 | **+2.01** | 60.80 | 39.25 | 37.44 |
| (w/ Spatial CoT & PointGraph) | 43.75 | **+2.86** | 62.79 | 38.64 | 37.74 |

In this section, we investigate the synergy between our two plug-and-play components, SpatialCoT and PointGraph. As shown in Table 10, across both GPT-4.1-mini and Qwen-VL2.5-3B, augmenting SpatialCoT with PointGraph yields an additional gain of roughly 1 percentage point in overall performance.

### F.2  FAILURE CASE ANALYSIS

Figure 8 illustrates a Spatial Interaction task instance from OmniSpatial, together with failure cases of two state-of-the-art commercial models, Gemini-2.5-Pro and ChatGPT-o3. We observe that, despite their strong capabilities, these models still struggle to reason over complex 3D scenes and to perform

orientation and path analysis under imagined egocentric poses. In this example, the model must first imagine itself standing in front of the washing machine and then plan a path in the 3D scene graph accordingly. We hypothesize that, if the model could generate intermediate visualizations of its planned route (e.g., hand-drawn trajectories) as a form of chain-of-thought (CoT), mimicking human problem-solving, it would be more effective at handling such challenging cases.

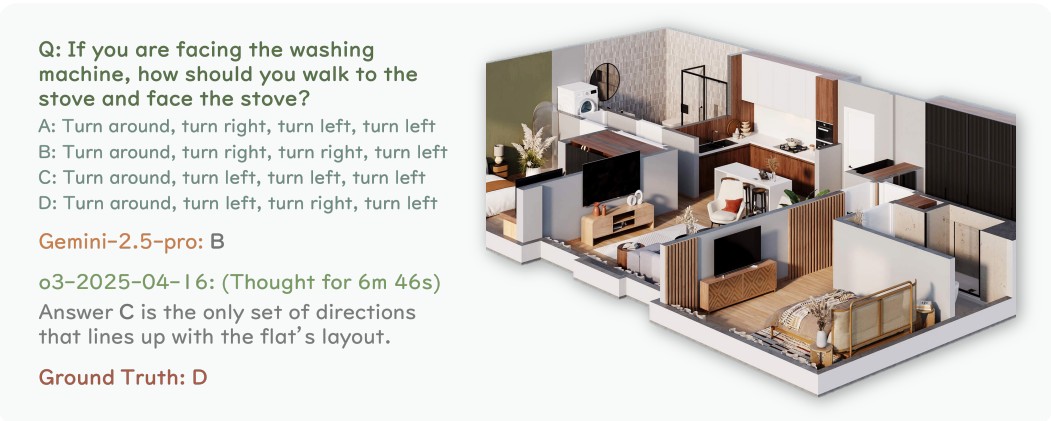

Figure 8: **Qualitative example for Failure Studies**.

In Fig. 9, we present a representative Perspective-Taking instance from OmniSpatial. We observe that even these top-performing models struggle with **frame-of-reference confusion**, counterfactual viewpoints and the associated spatial relations. In this example, the model must imagine itself facing the vase and then infer the relative positions of objects under this imagined viewpoint. Because current models find it difficult to construct and manipulate such fictitious perspectives, we propose using novel-view synthesis as a plug-and-play SpatialCoT module to enhance their Perspective-Taking ability. In future work, more tightly integrating this capability into the VLM's internal thinking process may further strengthen the model's spatial understanding.

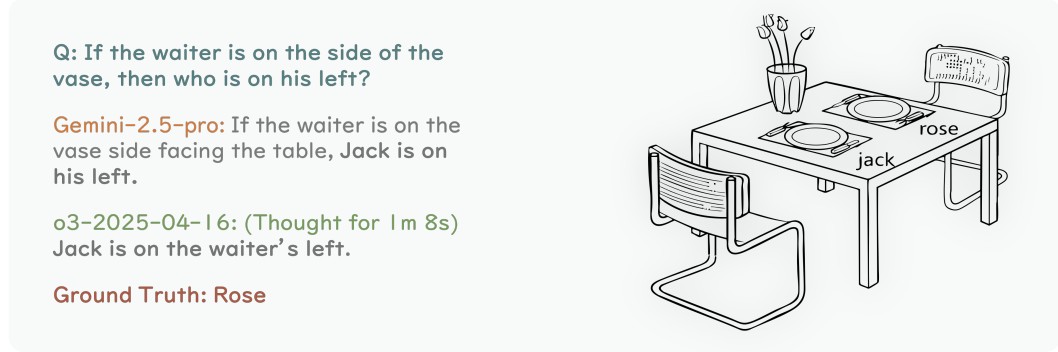

Figure 9: **Qualitative example for Failure Studies**.

In Fig. 10, we present an example spatio-temporal dynamics reasoning task from OmniSpatial. We observe that even state-of-the-art commercial models struggle to understand long-horizon sequences involving multiple continuous actions. We hypothesize that performance on such long-term temporal understanding tasks can be improved by augmenting the model's memory, for example by enriching its temporal and episodic representations of time–space relations.

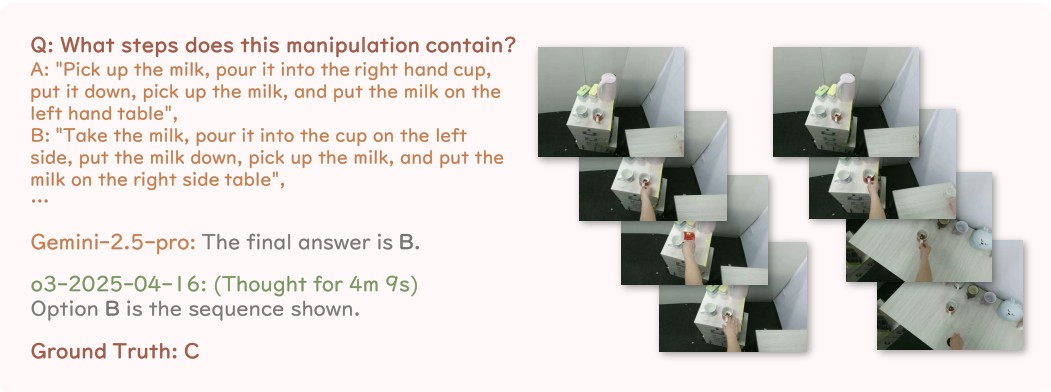

Figure 10: **Qualitative example for Failure Studies**.

In Figure 11, we present an example of a Complex-Logic task from OmniSpatial. In this problem, the model must understand the folding relationships in a paper-cutting process, which is highly challenging even for humans. We observe that the reasoning model produces a detailed chain of analysis, including examining how each folded corner affects the boundary and attempting to reason using structured text. However, we argue that such complex logical reasoning is better addressed with a vision-grounded chain-of-thought (CoT), i.e., a unified generative–perceptual model whose generative capability is used to predict the intermediate folded states step by step, gradually approaching the final correct prediction.

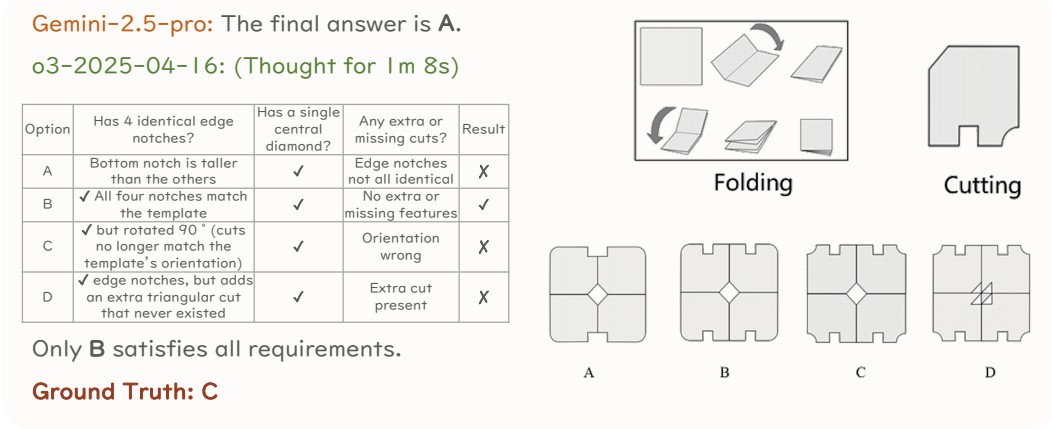

Figure 11: **Qualitative example for Failure Studies**.

In addition, we further conduct a quantitative analysis of how depth and object orientation affect the model's accuracy. We collect all real-image samples from the viewpoint conversion task and analyze them using Gemini-Flash-2.5 Reid et al. (2024). Object depth is estimated with Metric3D v2 Yin et al. (2023) and discretized into several ranges, while object orientation is evaluated with SoFar Qi et al. (2025) and grouped by angle. As shown in Fig. 12, we observe that the model's accuracy slightly decreases as the distance between the object and the camera increases. Regarding orientation, the model handles oblique views better, whereas its understanding of rear-view objects is somewhat weaker.

### F.3 MAIN RESULT WITH STANDARD DEVIATION

In Table 11, we additionally report results with standard deviations. Due to space limitations, we only present the means and standard deviations of the four coarse-grained categories, keeping the

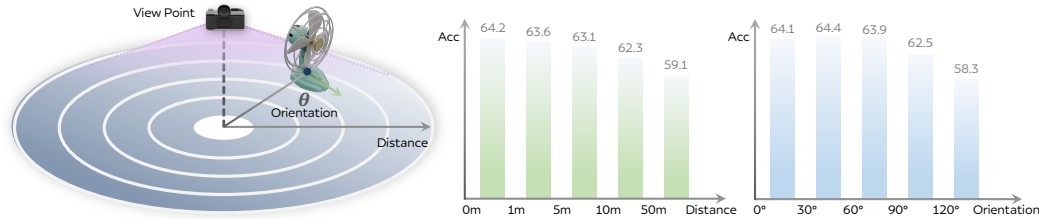

Figure 12: **Quantitative example for Failure Studies**.

Table 11: **Evaluation on OmniSpatial-test**. All models were tested 5 times and averaged to reduce randomness. We report accuracy on four coarse-grained categories: Dynamic Reasoning, Spatial Interaction, Complex Logic, and Perspective Taking. Each coarse category is the arithmetic mean of its corresponding fine-grained sub-tasks.

| Method | Avg. | Rank | Dynamic Reasoning | Spatial Interaction | Complex Logic | Perspective Taking |
|---|---|---|---|---|---|---|
| **Proprietary Models** | | | | | | |
| GPT-4o-mini-2024-07-18 (Hurst et al., 2024) | 42.64 ± 0.9 | 8 | 53.12 ± 1.1 | 47.64 ± 1.4 | 25.95 ± 0.4 | 44.18 ± 1.6 |
| GPT-4o-2024-11-20 (Hurst et al., 2024) | 47.81 ± 1.0 | 5 | 61.39 ± 1.3 | 54.31 ± 1.6 | 25.88 ± 0.5 | **51.74 ± 1.8** |
| GPT-4.1-nano-2025-04-14 (OpenAI, 2025a) | 42.62 ± 0.8 | 9 | 52.38 ± 1.0 | 46.09 ± 1.2 | 27.25 ± 0.4 | 41.52 ± 1.4 |
| GPT-4.1-mini-2025-04-14 (OpenAI, 2025a) | 48.87 ± 1.0 | 4 | 60.42 ± 1.2 | 58.54 ± 1.5 | 29.73 ± 0.5 | 50.47 ± 1.9 |
| GPT-4.1-2025-04-14 (OpenAI, 2025a) | 51.78 ± 1.1 | 2 | **65.48 ± 1.4** | 61.84 ± 1.7 | 30.91 ± 0.6 | 50.22 ± 1.7 |
| Claude-3-5-sonnet-20241022 (Anthropic, 2024) | 46.86 ± 0.9 | 6 | 54.31 ± 1.1 | 59.86 ± 1.3 | 29.17 ± 0.4 | 48.10 ± 1.5 |
| Claude-3-7-sonnet-20250219 (Anthropic, 2024) | 47.53 ± 0.9 | 5 | 56.76 ± 1.2 | 59.87 ± 1.4 | 28.94 ± 0.5 | 48.28 ± 1.6 |
| Gemini-2.0-flash-lite-02-05 (Anil et al., 2023) | 44.03 ± 0.8 | 8 | 52.95 ± 1.0 | 54.34 ± 1.2 | 26.44 ± 0.4 | 47.36 ± 1.4 |
| Gemini-2.0-flash-exp (Anil et al., 2023) | 48.40 ± 1.0 | 2 | 58.95 ± 1.3 | 58.09 ± 1.6 | 27.32 ± 0.5 | 50.41 ± 1.8 |
| Gemini-2.5-flash-preview-05-20 (Anil et al., 2023) | 52.12 ± 1.1 | 1 | 65.14 ± 1.5 | **67.49 ± 1.7** | **36.16 ± 0.6** | 48.37 ± 1.9 |
| **Reasoning Models** | | | | | | |
| o1-2024-12-17 (Jaech et al., 2024) | 50.36 ± 1.1 | 6 | 66.30 ± 1.1 | 60.49 ± 1.3 | 33.14 ± 1.3 | 48.58 ± 1.7 |
| o4-mini-2025-04-16 (OpenAI, 2025b) | 52.77 ± 1.2 | 3 | 66.40 ± 1.3 | 65.05 ± 1.5 | 35.40 ± 1.5 | 51.73 ± 1.9 |
| o3-2025-04-16 (OpenAI, 2025b) | 56.33 ± 1.3 | 1 | 69.03 ± 1.4 | 65.07 ± 1.6 | 34.95 ± 1.7 | **57.88 ± 2.0** |
| Claude-3-7-sonnet-20250219-thinking (Anthropic, 2024) | 48.62 ± 1.0 | 7 | 58.47 ± 1.0 | 59.65 ± 1.2 | 29.20 ± 1.1 | 47.84 ± 1.5 |
| Gemini-2.5-flash-05-20-thinking (Anil et al., 2023) | 53.16 ± 1.2 | 3 | 67.50 ± 1.2 | 63.91 ± 1.4 | 35.59 ± 1.4 | 49.20 ± 1.8 |
| Gemini-2.5-pro-preview-05-06 (Anil et al., 2023) | 55.19 ± 1.4 | 2 | **69.48 ± 1.5** | 67.38 ± 1.7 | **39.07 ± 1.8** | 49.96 ± 2.0 |
| **Open-source Models** | | | | | | |
| LLavA-1.5-vicuna-7B (Liu et al., 2024c) | 34.97 ± 0.7 | 15 | 42.84 ± 0.9 | 35.14 ± 1.1 | 26.59 ± 0.4 | 42.13 ± 1.3 |
| LLaVA-onevision-qwen2-7B (Li et al., 2024a) | 35.68 ± 0.7 | 14 | 40.70 ± 0.8 | 34.76 ± 1.0 | 25.73 ± 0.3 | 40.19 ± 1.2 |
| LLaVA-onevision-qwen2-72B (Li et al., 2024a) | 45.66 ± 0.9 | 6 | 56.22 ± 1.1 | 57.14 ± 1.3 | 24.24 ± 0.5 | 49.14 ± 1.6 |
| Gemma-3-4B (Kamath et al., 2025) | 39.79 ± 0.8 | 11 | 45.80 ± 0.9 | 40.15 ± 1.2 | 24.12 ± 0.4 | 44.84 ± 1.4 |
| Gemma-3-12B (Kamath et al., 2025) | 43.71 ± 0.8 | 8 | 54.48 ± 1.0 | 49.06 ± 1.3 | 23.41 ± 0.4 | 44.72 ± 1.5 |
| Gemma-3-27B (Kamath et al., 2025) | 44.75 ± 0.9 | 7 | 56.27 ± 1.1 | 53.62 ± 1.4 | 28.44 ± 0.5 | 43.58 ± 1.6 |
| InternVL3-2B (Zhu et al., 2025) | 37.98 ± 0.7 | 13 | 45.29 ± 0.8 | 41.28 ± 1.1 | 25.19 ± 0.3 | 41.20 ± 1.3 |
| InternVL3-8B (Zhu et al., 2025) | 41.60 ± 0.8 | 9 | 46.65 ± 0.9 | 48.25 ± 1.2 | 26.79 ± 0.4 | 47.93 ± 1.4 |
| InternVL3-14B (Zhu et al., 2025) | 45.94 ± 0.8 | 5 | 57.25 ± 1.0 | 51.20 ± 1.3 | 28.15 ± 0.4 | 45.96 ± 1.5 |
| InternVL3-38B (Zhu et al., 2025) | 48.48 ± 1.0 | 2 | **63.50 ± 1.4** | 54.48 ± 1.4 | 29.21 ± 0.5 | 47.47 ± 1.7 |
| InternVL3-78B (Zhu et al., 2025) | 49.33 ± 1.0 | 1 | 63.45 ± 1.3 | 55.64 ± 1.5 | 28.91 ± 0.5 | **49.62 ± 1.8** |
| Qwen-VL2.5-3B (Wang et al., 2024c) | 40.30 ± 0.8 | 10 | 51.46 ± 1.0 | 44.38 ± 1.2 | 28.02 ± 0.4 | 41.18 ± 1.4 |
| Qwen-VL2.5-7B (Wang et al., 2024c) | 39.18 ± 0.9 | 12 | 46.73 ± 1.0 | 46.48 ± 1.3 | **30.27 ± 0.6** | 45.02 ± 1.5 |
| Qwen-VL2.5-32B (Wang et al., 2024c) | 47.36 ± 1.0 | 4 | 59.08 ± 1.2 | **58.32 ± 1.5** | 26.94 ± 0.5 | 48.59 ± 1.7 |
| Qwen-VL2.5-72B (Wang et al., 2024c) | 47.85 ± 0.9 | 3 | 59.25 ± 1.2 | 54.52 ± 1.4 | 29.61 ± 0.5 | 48.19 ± 1.6 |
| **Specialized Spatial Reasoning Models** | | | | | | |
| SpaceMantis-13B (Chen et al., 2024) | 36.36 ± 0.7 | 6 | 41.81 ± 0.9 | 36.30 ± 1.1 | 23.33 ± 0.4 | 42.25 ± 1.3 |
| SpaceQwen2.5-VL-3B (Chen et al., 2024) | 40.25 ± 0.9 | 3 | 49.00 ± 1.0 | 41.01 ± 1.2 | **27.85 ± 0.5** | 47.44 ± 1.6 |
| SpaceThinker-Qwen2.5VL-3B (Chen et al., 2024) | 40.42 ± 0.9 | 2 | 50.45 ± 1.1 | 39.15 ± 1.3 | 26.16 ± 0.5 | 41.41 ± 1.4 |
| SpatialBot-3B (Cai et al., 2024) | 35.68 ± 0.7 | 6 | 40.70 ± 0.8 | 34.76 ± 1.0 | 25.73 ± 0.4 | 40.19 ± 1.2 |
| RoboPoint-vicuna-v1.5-7B-lora (Yuan et al., 2024) | 35.85 ± 0.7 | 6 | 42.82 ± 0.9 | 37.57 ± 1.1 | 26.30 ± 0.4 | 43.29 ± 1.3 |
| RoboPoint-vicuna-v1.5-13B (Yuan et al., 2024) | 34.60 ± 0.7 | 5 | 41.91 ± 0.9 | 35.85 ± 1.1 | 25.93 ± 0.4 | 40.06 ± 1.3 |
| SoFar-Qwen2.5VL-3B (Qi et al., 2025) | 45.14 ± 1.0 | 1 | **53.83 ± 1.2** | **53.33 ± 1.4** | 27.31 ± 0.6 | **49.95 ± 1.7** |
| **Human Evaluation** | | | | | | |
| Human | **92.63 ± 2.5** | - | **95.24 ± 1.3** | **93.48 ± 1.6** | **87.86 ± 5.5** | **94.36 ± 2.2** |

configuration consistent with the main table in the paper. We observe that the reasoning models exhibit larger variability across the five runs, whereas the smaller-parameter open-source models produce more consistent results.

## F.4 SPATIALCOT ON OTHER TRACK

Besides the perspective-taking tasks, SpatialCoT also improves performance on many problems that require viewpoint transformation. We further evaluate SpatialCoT on the Complex-Logic track; as shown in Table 12, SpatialCoT consistently achieves better performance than Manual CoT.

Table 12: **Performance of Spatial CoT on Complex-Logic track.**

| Method | Avg. | Improve | Pattern Recognition | Geometric Reasoning |
|---|---|---|---|---|
| **GPT-2.5-Flash-Thinking** | – | – | – | – |
| (w/ Manual CoT) | 35.59 | - | 35.05 | 36.13 |
| (w/ Spatial CoT) | 37.19 | **+1.60** | 35.30 | 38.28 |
| **Qwen-VL2.5-3B** | – | – | – | – |
| (w/ Manual CoT) | 28.02 | - | 32.16 | 23.87 |
| (w/ Spatial CoT) | 29.27 | **+1.25** | 29.01 | 28.52 |

## G  LIMITATION & FUTURE WORKS

Although OmniSpatial includes some image clips with dynamic information from HOI4D (Liu et al., 2022), the complexity of the operational tasks still lags behind that of long videos (Yang et al., 2024; Chandrasegaran et al., 2024). Moreover, while PointGraph & Spatial CoT enhances VLM's spatial understanding through point cues, the improvement is not fundamental in nature. Spatial reasoning tasks are more like mathematics and coding tasks, require longer and more complex reasoning (DeepSeek-AI et al., 2025; Jaech et al., 2024).

3D information is crucial for spatial reasoning, and future work involves introducing 3D representation (Dong et al., 2023; Qi et al., 2023a; 2024; 2023b) and perception (Qi et al., 2017a;b; Zhang et al., 2024a; Fan et al., 2024), as well as 3D VLMs (Peng et al., 2024; Dong et al., 2024; Guo et al., 2023; Qi et al., 2024; 2025), reasoning model (Zhang et al., 2025a; DeepSeek-AI et al., 2025; OpenAI, 2025b) and knowledge distillation (Zhang et al., 2023; Hinton et al., 2015; Dong et al., 2023; Qi et al., 2023a). The ultimate goal of spatial reasoning is to empower robots, and future work also involves robot execution tasks (Huang et al., 2024a; Fang et al., 2024a; Qi et al., 2025; Huang et al., 2024b; He et al., 2025b).

## H  BROADER IMPACTS

OmniSpatial promises several positive societal benefits. By pushing models to reason about motion, collision risk and traffic scenes, it can hasten the arrival of safer autonomous vehicles and service robots that foresee hazards and navigate crowded spaces responsibly. Its geometric-reasoning tasks—from polyhedron unfolding to assembly—offer data that can streamline product design, packaging and manufacturing, lowering material use and energy waste. The benchmark's localization, UI-interaction and perspective-taking challenges cultivate spatially aware assistants that improve AR/VR experiences, access tools for visually impaired users and more natural human-computer interfaces.

**[Default System Prompt]**

You are a spatial-reasoning assistant.

Task
-----
You will receive
1. **Image** - a single RGB frame depicting a scene.
2. **Question** - a natural-language query about spatial relationships between objects in the image.
3. **Options** - ≥2 answer candidates, each tagged by a capital letter (A, B, C, D…).

Based on the image and question, provide your answer.
Always ground your answer in the visual evidence; do not hallucinate unseen objects.
If uncertain, pick the most plausible option—never refuse or reply "insufficient information."

**[Zero-shot CoT System Prompt]**

You are a spatial-reasoning assistant.

Task
-----
You will receive
1. **Image** - a single RGB frame depicting a scene.
2. **Question** - a natural-language query about spatial relationships between objects in the image.
3. **Options** - ≥2 answer candidates, each tagged by a capital letter (A, B, C, D…).

Think step by step and provide the answer.
Always ground your answer in the visual evidence; do not hallucinate unseen objects.
If uncertain, pick the most plausible option—never refuse or reply "insufficient information."

**[Manual CoT System Prompt]**

You are a spatial-reasoning assistant.

Task
-----
You will receive
1. **Image** - a single RGB frame depicting a scene.
2. **Question** - a natural-language query about spatial relationships between objects in the image.
3. **Options** - ≥2 answer candidates, each tagged by a capital letter (A, B, C, D…).

Guidelines
----------
Please follow these steps to analyze the image and answer the question:
1. First, carefully observe the image and identify all relevant objects and their spatial relationships.
2. Next, break down the question into key components that need to be addressed.
3. Think through the spatial reasoning step-by-step to arrive at your answer. It may be necessary to transfer perspective to better understand the scene.
4. Finally, select the most appropriate option (A, B, C, or D) based on your analysis.

Always ground your answer in the visual evidence; do not hallucinate unseen objects.
If uncertain, pick the most plausible option—never refuse or reply "insufficient information."

Figure 13: **System prompts used in OmniSpatial evaluation.**

**[Blind Evaluation System Prompt]**

You are a spatial-reasoning assistant.

Task
-----
You will receive
1. **Question** - a natural-language query about spatial relationships.
2. **Options** - ≥2 answer candidates, each tagged by a capital letter (A, B, C, D...).

Based on the question only, provide your answer.

**[LLM Judgement System Prompt]**

You are a judge for QA tests.

The user will provide:
Question: The original question.
Pred: The predicted answer.
GT: The ground truth answer.

You need to judge whether the predicted answer is correct or not; just judge the final answer.
If the predicted answer is correct, respond with "True".
If the predicted answer is incorrect, respond with "False".

**[Direct System Prompt]**

You are a spatial-reasoning assistant.

Task
-----
You will receive
1. **Image** - a single RGB frame depicting a scene.
2. **Question** - a natural-language query about spatial relationships between objects in the image.
3. **Options** - ≥2 answer candidates, each tagged by a capital letter (A, B, C, D...).

Note: You only need to respond with A, B, C, or D without providing any additional information.

**[RE Format]**

End your answer with a separate line formatted exactly as:

Answer: X
where X ∈ {A, B, C, D}.

**[JSON Format]**

You need to respond with the answer in JSON format:

```json
{
"analysis": "The analysis of the image and question",
"answer": "A"
}
```

**[LLM Format]**
Your answer must be clear and accurate.

Figure 14: **System prompts used in OmniSpatial evaluation.**

**[Subtask Type: Motion Analysis]**

**[Task Type: Dynamic Reasoning]**

[**Question**]: If one of the three women in kimono walked toward the traffic light at 1 m/s, how long would it take her to reach it?

**[A]:** "13s"
**[B]:** "1.9s"
**[C]:** "12.7s"
**[D]:** "26.5s"

**[Answer]:** D

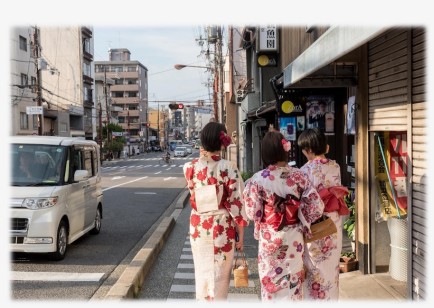

**[Subtask Type: Manipulation]**

**[Task Type: Dynamic Reasoning]**

[**Question**]: If it takes you 3 seconds to fully open the drawer, what is your average drawer opening speed?

**[A]:** "4.33inches/s"
**[B]:** "9.4inches/s"
**[C]:** "16.3inches/s"
**[D]:** "25.7inches/s "

**[Answer]:** A

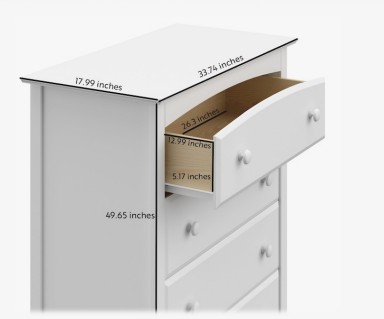

**[Subtask Type: Spatial Compatibility]**

**[Task Type: Dynamic Reasoning]**

[**Question**]: Is the gap between the construction vehicle and the cement container at least 0.6 meters wide, enough for a person to pass through?

**[A]:** "Yes"
**[B]:** "No"

**[Answer]:** A

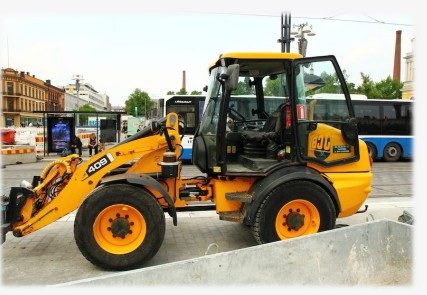

**[Subtask Type: Allocentiric]**

**[Task Type: Perspective Taking]**

[**Question**]: Which direction are they facing, from their own perspective?

**[A]:** "their left"
**[B]:** "their right"
**[C]:** "forward"
**[D]:** "backward"

**[Answer]:** B

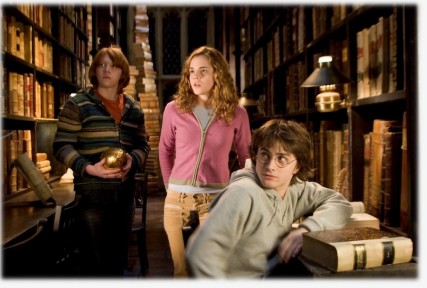

Figure 15: **Visualization example of OmniSpatial data samples.**

**[Subtask Type: Hypothetical]**

**[Task Type: Perspective Taking]**

[**Question**]: If you are riding on the tricycle, which direction is the white electric car from your perspective?

**[A]:** "left"
**[B]:** "right"
**[C]:** "forward"
**[D]:** "backward"

**[Answer]:** B

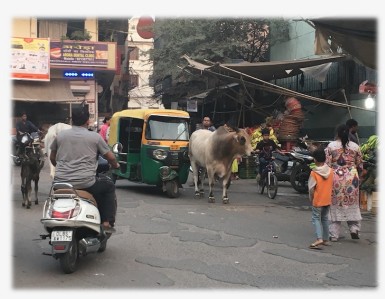

**[Subtask Type: Egocentric]**

**[Task Type: Perspective Taking]**

[**Question**]: How many dancers have their left foot in front?

**[A]:** "0"
**[B]:** "10"

**[Answer]:** A

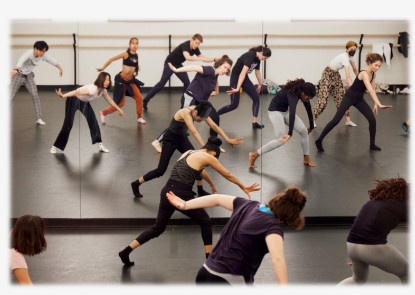

**[Subtask Type: Traffic Analysis]**

**[Task Type: Spatial Interaction]**

[**Question**]: I am driving on the highway. The safety distance is sufficient.

**[A]:** "Yes"
**[B]:** "No"

**[Answer]:** B

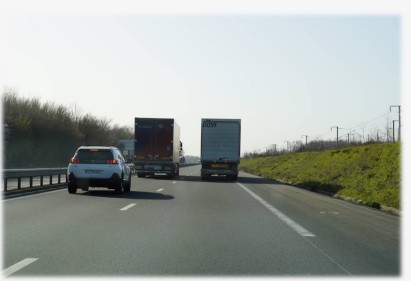

**[Subtask Type: Traffic Analysis]**

**[Task Type: Spatial Interaction]**

[**Question**]: While driving forward, what potential danger should you be aware of?

**[A]:** "0"
**[B]:** "1"
**[C]:** "2"
**[D]:** "3"

**[Answer]:** A

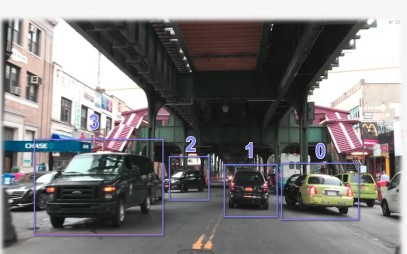

Figure 16: **Visualization example of OmniSpatial data samples.**

**[Subtask Type: Localization]**

**[Task Type: Spatial Interaction]**

[**Question**]: In case of an emergency, like a fall, what should you press on this interface?

**[A]:** "Click the first button from the left"
**[B]:** "Click the second button from the left"
**[C]:** "Click the third button from the left"
**[D]:** "Can not determine"

**[Answer]:** B

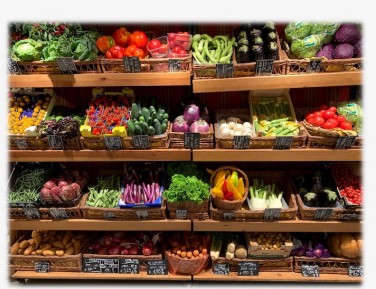

**[Subtask Type: Localization]**

**[Task Type: Spatial Interaction]**

[**Question**]: Where are the colored sweet peppers located on the shelf?

**[A]:** "Count the fourth line from the top and the first one from the right"
**[B]:** "Count the first line from the top and the second one from the right"
**[C]:** "Count the second line from the top and the third one from the right"
**[D]:** "Count the third line from the top and the fourth one from the right"

**[Answer]:** D

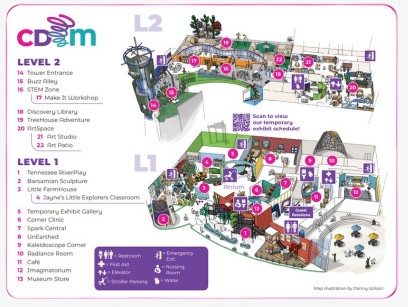

**[Subtask Type: Geospatial Strategy]**

**[Task Type: Spatial Interaction]**

[**Question**]: How can I get from the coffee shop on the first floor to the second floor?

**[A]:** "Go north and take the elevator"
**[B]:** "Go south and take the elevator"
**[C]:** "Go west and take the elevator"
**[D]:** "Go east and take the elevator"

**[Answer]:** D

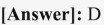

**[Subtask Type: Geospatial Strategy]**

**[Task Type: Spatial Interaction]**

[**Question**]: Which route utilizes the road more to support the run?

**[A]:** "green route"
**[B]:** "red route"
**[C]:** "purple route"
**[D]:** "orange route"

**[Answer]:** A

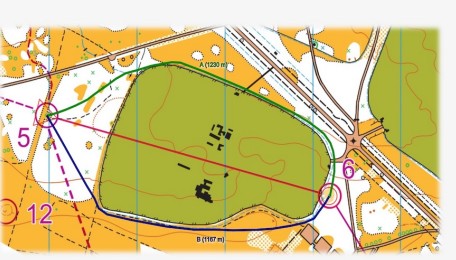

Figure 17: **Visualization example of OmniSpatial data samples.**

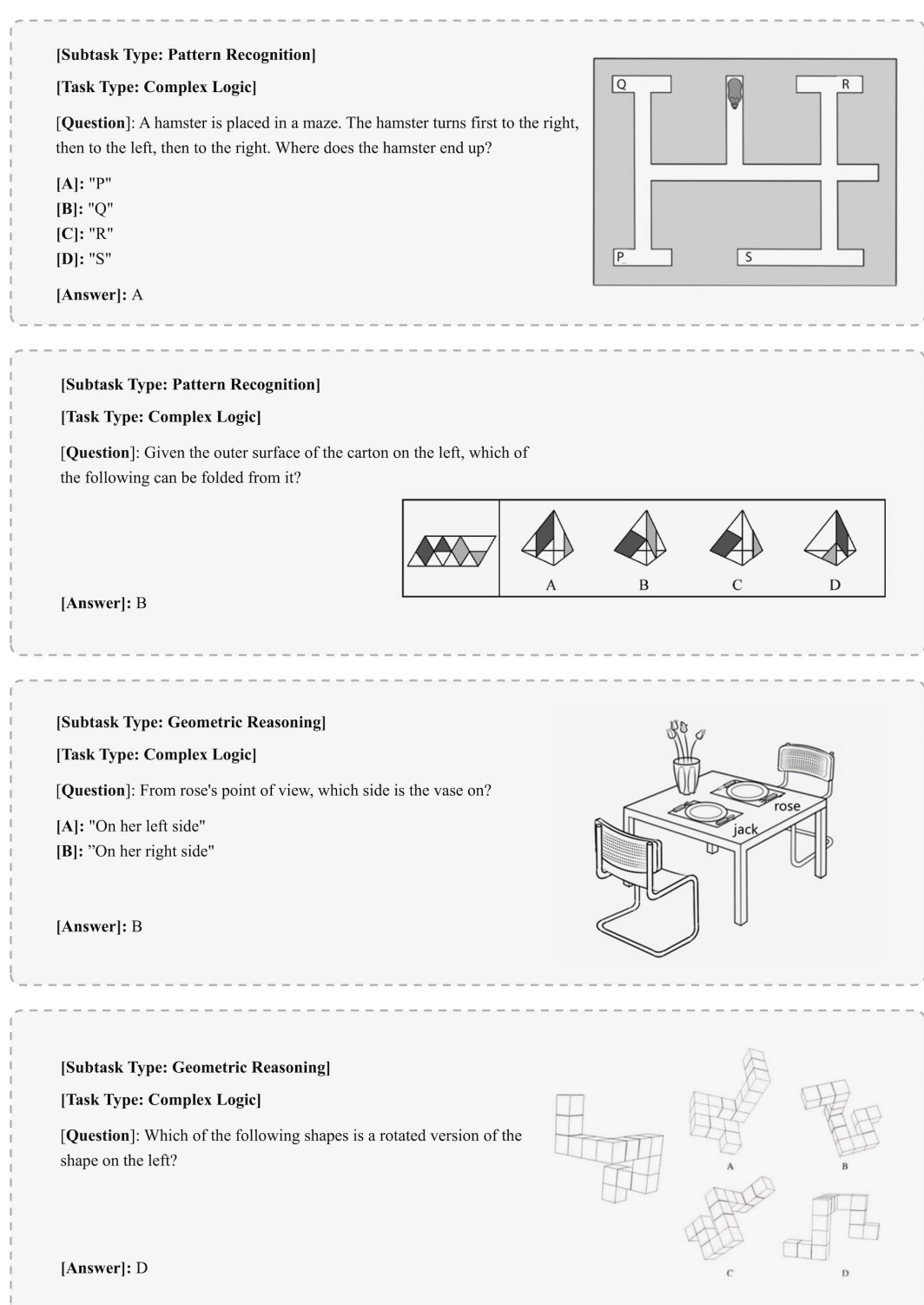

**[Subtask Type: Pattern Recognition]**

**[Task Type: Complex Logic]**

[**Question**]: A hamster is placed in a maze. The hamster turns first to the right, then to the left, then to the right. Where does the hamster end up?

**[A]:** "P"
**[B]:** "Q"
**[C]:** "R"
**[D]:** "S"

**[Answer]:** A

**[Subtask Type: Pattern Recognition]**

**[Task Type: Complex Logic]**

[**Question**]: Given the outer surface of the carton on the left, which of the following can be folded from it?

**[Answer]:** B

**[Subtask Type: Geometric Reasoning]**

**[Task Type: Complex Logic]**

[**Question**]: From rose's point of view, which side is the vase on?

**[A]:** "On her left side"
**[B]:** "On her right side"

**[Answer]:** B

**[Subtask Type: Geometric Reasoning]**

**[Task Type: Complex Logic]**

[**Question**]: Which of the following shapes is a rotated version of the shape on the left?

**[Answer]:** D

Figure 18: **Visualization example of OmniSpatial data samples.**

