# OpenReview forum: "OmniSpatial: Towards Comprehensive Spatial Reasoning Benchmark for Vision Language Models"
_ICLR.cc/2026/Conference — ICLR 2026 Poster_

### Official Review · Reviewer_EAe1 · 2025-10-28

**Soundness:** 3
**Presentation:** 3
**Contribution:** 3
**Rating:** 6
**Confidence:** 4

**Summary:**

This paper introduces OmniSpatial, a comprehensive and challenging benchmark designed to evaluate the spatial reasoning of VLMs. Grounded in cognitive psychology, OmniSpatial features over 8.4K manually curated question-answer pairs across four key categories: dynamic reasoning, complex spatial logic, spatial interaction, and perspective-taking. Experiments show that even SOTA VLMs struggle significantly, performing far below human accuracy. To bridge this gap, authors propose two novel strategies PointGraph and SpatialCoT, which leverage structured scene graphs and multi-view synthesis to improve the model’s reasoning capabilities.

**Strengths:**

This paper’s most significant contribution is exposing that top AI models fail at complex spatial reasoning where humans excel, clearly defining a crucial and challenging direction for future VLM research.
It presents the highly original concept of SpatialCoT, which enhances reasoning by simulating mental imagery. This creative fusion of 3D novel-view synthesis with chain-of-thought prompting represents a significant conceptual advance for tackling view-dependent and perspective-taking tasks.
The work is distinguished by its quality, evident in the meticulous manual creation of its 8.4K question-answer pairs, which achieved a high inter-annotator agreement and transparent evaluation across a wide spectrum of leading AI models.

**Weaknesses:**

The paper demonstrates that models fail on complex tasks but does not offer a deep analysis of the reasons. Without a breakdown of specific error types, the work provides limited actionable guidance for researchers to develop targeted architectural improvements.

**Questions:**

Given the performance gap between frontier models and humans, it's important to consider whether current methods can help VLMs catch up. If not, what future research or scaling approaches could bridge this gap?

---

> ### Author Response · Authors · 2025-11-21
>
> Thank you very much for your positive assessment of our work and for the insightful suggestions. We address your concerns in detail below.
>
> **[W1] Lack of qualitative examples for Failure Studies**
>
> Thank you for this valuable suggestion. In the revised version, we have added a new subsection Failure Case Analysis in the appendix F.2. There, we provide qualitative and quantitative examples that systematically examine typical failure modes. Our analysis confirms that current VLMs still exhibit notable limitations in depth perception, orientation understanding, frame-of-reference reasoning, and temporal alignment, especially in complex spatial scenes.
>
> **[Q1] Future research direction**
>
> This is an excellent and important question. We see several promising directions that may advance the spatial intelligence of VLMs:
>
> 1. **3D/4D representations and encodings.**
> Spatial intelligence is inherently about reasoning in a 3D/4D world. Incorporating 3D/4D representations, enforcing 3D-aware understanding, or injecting 3D positional encodings into VLMs could substantially enhance their spatial reasoning capabilities.
>
> 2. **Leveraging grounding and segmentation as spatial primitives.**
> Tasks such as visual grounding and segmentation provide the basic primitives for spatial relations. We believe that exploiting a VLM’s perception and grounding abilities—either as auxiliary objectives or as a form of spatial chain-of-thought (CoT)—can further strengthen its spatial reasoning.
>
> 3. **Ego-centric learning of spatial relations.**
> Humans acquire spatial concepts through long-term ego-centric visual experience. Similarly, promoting ego-centric understanding and training (e.g., first-person trajectories, viewpoint changes, and self-centered coordinates) may significantly improve a model’s ability to reason about both self-centered and relative perspectives.
>
> 4. **Spatio-temporal memory for persistent spatial understanding.**
> Spatio-temporal understanding is tightly coupled with memory. Enabling VLMs to maintain structured episodic and semantic memories over time—especially about time–space relationships—could help them build more consistent and robust spatial representations, thereby improving their spatial reasoning ability in dynamic scenes.
>
>
> **[Flag for Ethics Review]**
>
> Additionally, we would like to clarify the status of the **Flag for Ethics Review**. Our benchmark does not involve any ethics-related risks: all annotations are based on detailed annotation guidelines and verification procedures, and have undergone multiple rounds of checks to mitigate potential biases. We therefore kindly request that the Flag for Ethics Review be set to “No.”

---

> > ### Comment · Reviewer_EAe1 · 2025-11-26
> >
> > Thanks for answering my questions and I will keep my rating.

---

### Official Review · Reviewer_QVvP · 2025-10-31

**Soundness:** 3
**Presentation:** 4
**Contribution:** 3
**Rating:** 8
**Confidence:** 4

**Summary:**

The paper proposes a new Visual Question Answering (VQA) benchmark, OmniSpatial, designed to comprehensively evaluate spatial reasoning capabilities. The benchmark highlights a broad and in-depth coverage of spatial relation tasks. The authors carefully design four key dimensions of spatial reasoning to be evaluated: perspective-taking, spatial interaction, dynamic reasoning, and compositional understanding. These four categories present significant challenges and substantially advance the complexity of spatial evaluation for current vision-language models (VLMs).
The proposed benchmark provides a comprehensive evaluation across diverse conditions, supported by a large collection of curated web images. The paper conducts experiments across multiple model series and sizes, including reasoning, closed-source, and open-source VLMs, while also providing a human baseline for comparison.
In addition, the authors investigate two approaches aimed at improving VLM spatial understanding on this benchmark, PointGraph and SpatialCoT. Both  yield consistent improvements across different VLMs.

**Strengths:**

- The proposed benchmark introduces a new and challenging evaluation setting that explores aspects of spatial reasoning rarely addressed in previous datasets. It is notably more complex and comprehensive than prior benchmarks.
- The question annotations involve a human-in-the-loop process to ensure clarity, answer uniqueness, and the resolution of ambiguous spatial references.
- The evaluation includes a wide range of VLMs—covering reasoning-focused, open-source, closed-source, and human baselines—demonstrating the benchmark’s broad coverage, thorough experimental setup, and comprehensive comparisons across models.
- Results demonstrate the significant shortcomings of VLMs across different type of spatial relation
- The paper also introduces two promising approaches, PointGraph (which incorporates an explicit scene graph as input) and SpatialCoT (which generates multi-view points from a given image to provide diverse spatial perspectives). These methods consistently improve model performance across different VLMs.
- Paper shows that fine-tuning models with this dataset shows potential for transferability to other VLM benchmarks.
- The paper is well-written and includes clear illustrations that help the audience understand the proposed spatial relation benchmark and its evaluation scope.

**Weaknesses:**

- There is no qualitative analysis of failure cases. Investigating these failures would strengthen the paper further. Providing a few examples and categorizing the errors could help reveal which aspects of reasoning need improvement—such as perception, logical reasoning, or consistency.
- The paper only demonstrates the effectiveness of SpatialCoT on the perspective-taking task. How does this approach affect performance on other task types? This might raise some concern that it make model perform worth in other tasks that does not require perspetive taking.
- Minor issue, Table 4 is never mentioned in discussion.

**Questions:**

- In the question generation process, is a fixed template used, or are LLM involved? If the process relies on template-based questions, would it be possible to incorporate LLMs to increase the diversity of question types? If it already use LLM, the paper should expicitly said so.

---

> ### Author Response · Authors · 2025-11-21
>
> We thank you for your positive assessment of our work and your insightful comments. Below we address each of your concerns in detail.
>
> **[W1] Lack of qualitative examples for Failure Studies**
>
> Thank you for this valuable suggestion. In the revised appendix F.2, we have added a new section titled Failure Case Analysis. This section provides both quantitative and qualitative examples of typical failure modes. Our analysis confirms that current VLMs still exhibit clear limitations in robust depth perception, orientation understanding, frame-of-reference consistency, and temporal alignment.
>
> **[W2] SpatialCoT on other tasks**
>
> We appreciate the opportunity to clarify the intended scope of SpatialCoT. SpatialCoT is specifically designed to mitigate VLM weaknesses in “mental imagery” and viewpoint transformation. For tasks that do not require such viewpoint changes (e.g., purely planar map understanding or temporal-dynamics analysis), we agree that SpatialCoT does not bring additional performance gains. However, for problems that involve spatial transformations and multi-step spatial reasoning, we observe consistent improvements when using SpatialCoT. We have added corresponding experiments and analyses for these tasks in the revised appendix.
>
> **[W3] Table 4 is never mentioned**
>
> Thank you for pointing out this oversight. In the revised manuscript, we have updated the corresponding section to explicitly reference and discuss Table 4, ensuring that its role and implications are clearly explained.
>
> **[Q1] In the question generation process, is a fixed template used, or are LLMs involved?**
>
> In the OmniSpatial test set, all questions are written and verified by human annotators. No fixed templates and no LLM-generated questions are used. This is a key difference between OmniSpatial and many existing spatial benchmarks. We intentionally avoid templates because they tend to impose rigid formats and may encourage models to overfit to superficial patterns, thereby undermining the benchmark’s ability to evaluate genuine spatial generalization.

---

### Official Review · Reviewer_8wJb · 2025-11-01

**Soundness:** 3
**Presentation:** 3
**Contribution:** 2
**Rating:** 4
**Confidence:** 4

**Summary:**

This paper presents OmniSpatial, a new and comprehensive benchmark aimed at evaluating higher-level spatial reasoning in VLMs beyond basic left–right or counting tasks. It provides 8.4K human-curated QA pairs across four categories—dynamic reasoning, complex spatial logic, spatial interaction, and perspective-taking—covering 50 task types. Evaluating 36 models shows that state-of-the-art VLMs achieve only 56% accuracy, far below human performance, with notable weaknesses in geometric reasoning and non-egocentric perspective shifts. The paper also introduces PointGraph and SpatialCoT as two strategies to improve spatial reasoning, both yielding modest gains.

**Strengths:**

- The paper is well written and easy to understand.
- The dataset construction is solid and carefully annotated by humans.
- The evaluation is comprehensive.

**Weaknesses:**

**Training Data Leakage Concern**
- While the dataset is manually curated, some sources (e.g., web images, exam-style questions) may overlap with model pretraining corpora. A clearer discussion on leakage mitigation, measurement, and dataset decontamination would strengthen the benchmark’s credibility.

**Compute Cost of SpatialCoT**
- The proposed SpatialCoT relies on multi-view synthesis, which appears computationally expensive. A discussion of its runtime, resource requirements, and potential lightweight or more practical alternatives would improve the clarity of its applicability.


**Lack of Discussion on Related Works**
-  I have seen prior works that also incorporate structured spatial information through text-based scene representations (e.g.,[1]). The PointGraph idea seems to be closely related to this one. It would be appropriate to acknowledge and discuss such related methods when introducing PointGraph in Sec. 3.3.1 to better position the contribution.

**Missing Error Bars in Reporting Results**
The main table does not present confidence intervals, variance, or statistical testing. As this is a benchmark paper, stronger evidence of robustness and significance is needed. Reporting standard deviations or significance tests can better support the claims and ensure results are reliable.

Overall, I believe the paper could be a good contribution to the community, and I would be happy to reconsider the score if the above concerns are satisfactorily addressed.

**Questions:**

- To what extent can models answer correctly without looking at the images? Since most questions are binary or 4-way multiple choice, some may be solvable from textual priors alone (e.g., Fig. 3: “I am entering a highway, I will encounter a ‘Give Way’ sign”). Have the authors evaluated a text-only baseline to isolate true visual reasoning?

- How is PointGraph different from existing methods like [1]?

- Can the authors provide an estimated compute overhead of SpatialCoT and discuss practicality for deployment?

- How do the authors assess or mitigate potential data leakage, especially for web- or exam-derived content that may exist in model training corpora? Is there any decontamination or measurement of overlap?

[1] Wang et al., Is A Picture Worth A Thousand Words? Delving Into Spatial Reasoning for Vision Language Models, NeurIPS 2024

---

> ### Author Response · Authors · 2025-11-20
>
> We sincerely thank you for your thoughtful comments on our paper. Below we address your concerns in detail, and we would be very grateful if you would consider adjusting your rating in light of these clarifications.
>
> **[W1 & Q4] Training Data Leakage Concern**
>
> Since we rely on commercial open- and closed-source models (e.g., ChatGPT, Gemini, Qwen), we do not have access to their exact pre-training corpora, nor can we precisely determine which Internet data or public/private datasets they were trained on. Fortunately, all questions and answers in OmniSpatial were manually annotated by our human annotators from scratch, without any assistance from LLMs or template-based generation. Therefore, OmniSpatial does not suffer from leakage of questions or answers from any LLM-generated or templated source.
>
> **[W2 & Q3] Compute Cost of SpatialCoT**
>
> For SpatialCoT, we use the multi-view generation model from InstantMesh as a chain-of-thought (CoT) module to enhance VLMs’ ability to reason about viewpoint changes. The multi-view generator in InstantMesh is obtained by fine-tuning Zero123++, which in turn is built on Stable Diffusion XL-1.0 as the backbone for novel-view synthesis. Stable Diffusion XL-1.0 has about 3.5B parameters, which is comparable in scale to many commonly used VLMs (e.g., Qwen-7B, InternVLM-32B) and not larger in order of magnitude.
>
> Moreover, our framework only calls this image generator once per inference to obtain the auxiliary views. In our measurements, Zero123++ requires only approximately 1.0 seconds to generate a single image on a single RTX 4090 GPU, which we consider a reasonable overhead for the gains in spatial understanding.
>
> **[W3 & Q2] Lack of Discussion on Related Works**
>
> We appreciate the pointer to related work. However, we would like to clarify that our proposed PointGraph is fundamentally different from SpatialEval [1].
>
> Task domain. SpatialEval focuses on spatial reasoning over abstract, symbolic patterns such as mazes and grids, where the inputs are already highly structured. In contrast, OmniSpatial contains rich real-world visual tasks, including spatiotemporal dynamic analysis, complex logical reasoning, human–robot interaction, and viewpoint transformation.
>
> Motivation for PointGraph. Through both qualitative and quantitative analysis, we observe that VLMs’ object localization and segmentation capabilities lag behind those of specialized detection/segmentation models (e.g., GroundedSAM, Florence-2). This motivates us to leverage such specialized models to build explicit scene graphs that serve as auxiliary signals to strengthen VLMs’ spatial reasoning.
>
> SpatialEval uses formatted textual symbols such as:
> ```
> #######
> ###S###
> ##S####
> #######
> ```
>
> In contrast, our PointGraph is constructed by first localizing and segmenting the referred objects with specialized models and then encoding them into a scene graph representation like the following, which is provided as auxiliary information to the VLM:
> ```json
> {
>   "apple": {
>     "center": [263, 391],
>     "bbox": [[242, 367], [284, 415], [242, 391], [284, 415]]
>   },
>   "robot gripper": {
>     "center": [765, 628],
>     "bbox": [[731, 584], [799, 672], [731, 584], [799, 672]]
>   }
> }
> ```
>
> Given the conceptual relevance, we have now explicitly cited SpatialEval [1] and discussed the relationship and key differences in the revised version.
>
> **[W4] Missing Error Bars in Reporting Results**
>
> Thank you for highlighting the importance of statistical robustness. In the revised version, we have added complete evaluation results with error bars (standard deviations) in the appendix. These extended tables cover all categories. In the main paper, we originally omitted them due to strict page limits and the large number of categories, but we agree that including them in the supplementary material improves the reliability and transparency of the benchmark.
>
> **[Q1] Have the authors evaluated a text-only baseline to isolate true visual reasoning?**
>
> Yes. In the initial submission, we already provided a blind evaluation in Table 2, including Random Choice, GPT-3.5-Turbo, and GPT-4-Turbo as text-only baselines. In these experiments, we remove all images and only provide the textual components (e.g., question text, answer options), thereby isolating the contribution of visual information.
>
> The results show that, on OmniSpatial, models without access to visual inputs perform very poorly, confirming that purely textual cues are insufficient and that our benchmark indeed requires genuine visual–spatial reasoning rather than exploiting dataset biases or textual shortcuts.

---

### Official Review · Reviewer_Qifc · 2025-11-01

**Soundness:** 3
**Presentation:** 3
**Contribution:** 3
**Rating:** 4
**Confidence:** 2

**Summary:**

This paper introduces OmniSpatial, a large-scale benchmark designed to evaluate comprehensive spatial reasoning in vision-language models (VLMs). It organizes tasks into four key categories, including dynamic reasoning, complex spatial logic, spatial interaction, and perspective-taking, covering 50 subtypes and 8.4K manually curated QA pairs. The benchmark integrates multiple data sources (web, cognitive tests, driving exams, and prior embodied datasets) with high annotation consistency (Krippendorff’s α = 0.84).

The authors further propose two methods to enhance VLM spatial reasoning:
1. PointGraph – providing explicit scene graphs for spatial structure.
2. SpatialCoT – enabling multi-view reasoning using novel-view synthesis (InstantMesh).

They benchmark 36 VLMs (GPT-4.1, Gemini-2.5, Qwen-VL, InternVL, etc.) and show that while leading reasoning models (e.g., o3, Gemini-2.5-pro) achieve ≈56% accuracy, human performance reaches 92%. Fine-tuning on OmniSpatial improves performance (+7.8 points) and transfers modestly to other spatial benchmarks (e.g., VSI-Bench +2 points)

**Strengths:**

1. Focused and Systematic Scope
The paper maintains a clear focus on spatial reasoning, defining it precisely, covering its cognitive dimensions, and avoiding unnecessary general multimodal extensions. This conceptual focus makes OmniSpatial a coherent and practically usable benchmark.

2. Rigorous Manual Curation:
- The dataset is human-annotated, multi-sourced, and cross-validated with strong inter-annotator agreement, addressing common weaknesses of synthetic or template-based datasets.

**Weaknesses:**

1. Lack of Deep Analysis or Failure Studies
The paper could benefit from qualitative examples showing why models fail (e.g., depth reasoning errors, frame-of-reference confusion, or temporal misalignment)

2. Marginal Quantitative Gains
The improvements from PointGraph and SpatialCoT are modest (≈1–2 points per dimension), raising questions about their practical impact.

**Questions:**

1. How do PointGraph and SpatialCoT interact? Are their improvements additive or overlapping?
2. Can the authors provide qualitative examples illustrating typical model errors (e.g., misinterpreting object orientation, inconsistent frame of reference)?

---

> ### Author Response · Authors · 2025-11-20
>
> We thank the reviewer for the insightful comments and constructive suggestions. Below we address each concern in detail. We hope our clarifications and new experiments resolve your concerns, and we would be very grateful if you could consider raising your score after reading this rebuttal.
>
> **[W1 & Q2] Lack of qualitative examples for Failure Studies**
>
> Thank you for pointing this out. In the revised version, we have added a new section “F.2 Failure Case Analysis” in the appendix. There, we provide both qualitative examples and quantitative analyses of typical failure modes. Specifically, we systematically study how depth perception, object orientation, frame-of-reference confusion, and temporal misalignment affect model predictions. Our analysis confirms that current VLMs indeed exhibit clear limitations along these dimensions, which we illustrate with concrete examples to make the failure patterns more interpretable and actionable for future work.
>
> **[W2] Marginal Quantitative Gains of PointGraph and SpatialCoT**
>
> PointGraph and SpatialCoT are explicitly motivated by the above failure analyses. PointGraph leverages scene graphs to strengthen relational reasoning, while SpatialCoT uses novel-view synthesis to improve robustness under viewpoint changes. Although the average gains are about 2–3 points, we emphasize that OmniSpatial is a highly challenging benchmark; such improvements are substantial and correspond roughly to the performance gap between, e.g., Gemini-2.5-Flash and Gemini-2.5-Pro or between GPT-o4-mini and GPT-o1 on our benchmark. Moreover, both PointGraph and SpatialCoT are designed as plug-and-play modules that can be easily applied to existing VLMs without re-training from scratch, providing a practical and general way to enhance spatial reasoning.
>
> **[Q1] Are PointGraph and SpatialCoT improvements additive or overlapping?**
>
> Conceptually, PointGraph and SpatialCoT target complementary aspects of spatial understanding: PointGraph focuses on improving relational reasoning via scene graphs, while SpatialCoT enhances viewpoint robustness via novel-view synthesis. We therefore expect their effects to be largely additive rather than overlapping. To validate this, we have included new experiments in Appendix F.1 of the revised version. The results show that combining both techniques consistently yields cumulative performance gains, supporting our claim that the two modules improve different, complementary facets of VLM spatial reasoning.

---

### Author Response · Authors · 2025-11-27
**general response**

We sincerely thank the reviewers and the AC for their time, constructive feedback, and thoughtful comments. We are encouraged that OmniSpatial was recognized as a “systematic” (Qifc) and “comprehensive” (8wJb, QVvP) benchmark that addresses a critical gap in spatial reasoning (EAe1). We also appreciate the reviewers praising the “rigorous manual curation” (Qifc, 8wJb) of the dataset and acknowledging that our proposed methods, PointGraph and SpatialCoT, are “creative” and “promising” (QVvP, EAe1). To address the reviewers’ concerns, we have conducted additional analyses and experiments to further strengthen the paper:

**1. New Section**: Failure Case Analysis: We added Appendix F.2 to provide the requested qualitative and quantitative analysis of failure modes (e.g., depth perception, frame-of-reference confusion), offering actionable insights for future research. (Qifc, QVvP, EAe1)

**2. Additional Experiments on Method Synergy**: We included new experiments (Appendix F.1) demonstrating that PointGraph and SpatialCoT yield additive performance gains, confirming they target complementary aspects of spatial reasoning. (Qifc)

**3. Enhanced Statistical Reporting**: We added standard deviations and error bars to the evaluation results (Appendix F.3) to ensure statistical robustness and significance. (8wJb)

**4. Clarifications on Related Work & Cost**: We explicitly discussed the distinction between PointGraph and SpatialEval, analyzed the compute overhead of SpatialCoT, and reaffirmed data integrity through our text-only baseline and human curation process. (8wJb, QVvP)

**5. Future directions**: a new discussion outlining research avenues to close the human–model gap, including 3D/4D representations, grounding/segmentation primitives, egocentric learning, and spatio-temporal memory. (EAe1)


**All revisions are marked in blue in the updated manuscript.**  Below, we respond to each reviewer’s specific comments. We welcome further discussion and greatly appreciate your support. Thank you!

---

### Meta-Review · Area_Chair_KSFb · 2025-12-21

**Summary:**

This paper introduces a new benchmark (OmniSpatial) for comprehensive spatial reasoning, noting that existing benchmarks have been saturated and that more practical spatial reasoning necessitates complex questions on perspective taking, dynamic reasoning, spatial interaction, and complex logic. The paper also introduces two strategies for improving reasoning.

The paper received two positive and two negative reviews. Multiple reviewers remarked on limited discussion of failure case analysis of existing models on OmniSpatial and guidance on how to improve. The authors introduced a new section in the Appendix with additional analysis including on failure cases to help clarify these points.

My own reading agrees with the reviewers initial concerns on providing a more concrete guidance on where current models fail and how to advance the state-of-the-art. I encourage the authors to revise the paper slightly to bring this crucial information into the main text rather than hidden in an Appendix.

**Reviewer Concerns:**

- Questions about qualitative failure analysis was resolved.
- Questions about compute cost and data leakage were clarified. The rebuttal clarified comparisons to a prior work, SpatialEval.
- Questions about evaluations with blind models were resolved.
- Questions about performance gains of PointGraph and SpatialCoT are moderately addressed.

**Reviewer Scores:**

Of the two positive reviews, one reviewer iterated that their score remained the same. Of the two negative reviews, I believe they may have raised their score slightly given the new failure analysis discussion.

---

### Decision · Program_Chairs · 2026-01-26

Accept (Poster)